# RECURRENT EVENT NETWORK: GLOBAL STRUCTURE INFERENCE OVER TEMPORAL KNOWLEDGE GRAPH

## ABSTRACT

Modeling dynamically-evolving, multi-relational graph data has received a surge of interests with the rapid growth of heterogeneous event data. However, predicting future events on such data requires global structure inference over time and the ability to integrate temporal and structural information, which are not yet well understood. We present Recurrent Event Network (RE-NET), a novel autoregressive architecture for modeling temporal sequences of multi-relational graphs (*e.g.*, temporal knowledge graph), which can perform sequential, global structure inference over future time stamps to predict new events. RE-NET employs a *recurrent event encoder* to model the temporally conditioned joint probability distribution for the event sequences, and equips the event encoder with a *neighborhood aggregator* for modeling the concurrent events within a time window associated with each entity. We apply teacher forcing for model training over historical data, and infer graph sequences over future time stamps by sampling from the learned joint distribution in a sequential manner. We evaluate the proposed method via temporal link prediction on five public datasets. Extensive experiments[1] demonstrate the strength of RE-NET, especially on multi-step inference over future time stamps.

## 1 INTRODUCTION

Representation learning on dynamically-evolving, graph-structured data has emerged as an important problem in a wide range of applications, including social network analysis (Zhou et al., 2018a; Trivedi et al., 2019), knowledge graph reasoning (Trivedi et al., 2017; Nguyen et al., 2018; Kazemi et al., 2019), event forecasting (Du et al., 2016), and recommender systems (Kumar et al., 2019; You et al., 2019). Previous methods over dynamic graphs mainly focus on learning time-sensitive structure representations for node classification and link prediction in single-relational graphs. However, the rapid growth of heterogeneous event data (Mahdisoltani et al., 2014; Boschee et al., 2015) has created new challenges on modeling temporal, complex interactions between entities (*i.e.*, viewed as a *temporal knowledge graph* or a TKG), and calls for approaches that can predict new events in different future time stamps based on the history—*i.e.*, structure inference of a TKG over time.

Recent attempts on learning over temporal knowledge graphs have focused on either predicting missing events (facts) for the observed time stamps (García-Durán et al., 2018; Dasgupta et al., 2018; Leblay & Chekol, 2018), or estimating the conditional probability of observing a future event using temporal point process (Trivedi et al., 2017; 2019). However, the former group of methods adopts an *interpolation* problem formulation over TKGs and thus cannot predict future events, as representations of unseen time stamps are unavailable. The latter group of methods, including Know-Evolve and its extension, DyRep, computes the probability of future events using ground-truths of the proceeding events during inference time, and cannot model concurrent events occurring within the same time window—which often happens when event time stamps are discrete. It is thus desirable to have a principled method that can infer graph structure sequentially over time and can incorporate local structural information (*e.g.*, concurrent events) during temporal modeling.

To this end, we propose a sequential structure inference architecture, called Recurrent Event Network (RE-NET), for modeling heterogeneous event data in the form of temporal knowledge graphs. Key ideas of RE-NET are based on the following observations: (1) predicting future events can be viewed as a sequential (multi-step) inference of multi-relational interactions between entities over time; (2)

---

[1]Code and data have been uploaded and will be published upon acceptance of the paper.

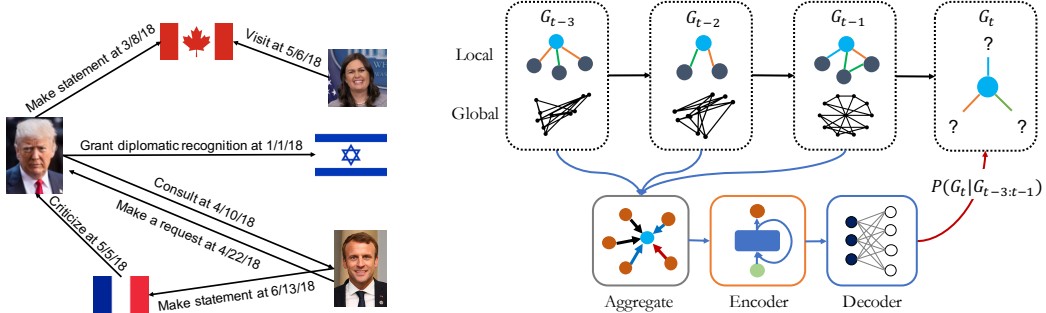

(a) An example of temporal KG      (b) Overview of the RE-NET architecture

Figure 1: **Illustration of (a) temporal knowledge graph and (b) the Recurrent Event Network architecture.** RE-NET employs an RNN to capture s-related interactions $N_t^{(s)}$ (modeled by a neighborhood aggregator) at different time $t$. Also the global information from $G_t$ is used to capture the global graph structures. Recurrent event encoder updates its state with graph sequences in an autoregressive manner. The decoder defines the probability $\mathbb{P}(s_t, r_t, o_t | G_{:t-1})$ at current time step conditioned on the preceding events.

temporally adjacent events may carry related semantics and informative patterns, which can further help inform future events (*i.e.*, temporal information); and (3) multiple events may co-occur within the same time window and exhibit structural dependencies as they share entities (*i.e.*, local structural information). To incorporate these ideas, RE-NET defines the joint probability distribution of all the events in a TKG in an autoregressive fashion, where it models the probability distribution of the concurrent events at the current time step conditioned on all the preceding events (see Fig. 1b for an illustration). Specifically, a recurrent event encoder, parametrized by RNNs, is used to summarize information of the past event sequences, and a neighborhood aggregator is employed to aggregate the information of concurrent events for the related entity within each time stamp. With the summarized information of the past event sequences, our decoder defines the joint probability of a current event. Such an autoregressive model can be effectively trained by using teacher forcing. Global structure inference for predicting future events can be achieved by performing sampling in a sequential manner.

We evaluate our proposed method on temporal link prediction task, by testing the performance of multi-step inference over time on five public temporal knowledge graph datasets. Experimental results demonstrate that RE-NET outperforms state-of-the-art models of both static and temporal graph reasoning, showing its better capacity to model temporal, multi-relational graph data with concurrent events. We further show that RE-NET can perform effective multi-step inference to predict unseen entity relationships in a distant future.

## 2 RELATED WORK

Our work is related to previous studies on temporal knowledge graph reasoning, temporal modeling on homogeneous graphs, recurrent graph neural networks, and deep autoregressive models.

**Temporal KG Reasoning.** There are some recent attempts on incorporating temporal information in modeling dynamic knowledge graphs. (Trivedi et al., 2017) presented Know-Evolve which models the occurrence of a fact as a temporal point process. However, this method is built on a problematic formulation when dealing with concurrent events, as shown in Section F. Several embedding-based methods have been proposed (García-Durán et al., 2018; Leblay & Chekol, 2018; Dasgupta et al., 2018) to model time information. They embed the associate into a low dimensional space such as relation embeddings with RNN on the text of time (García-Durán et al., 2018), time embeddings (Leblay & Chekol, 2018), and temporal hyperplanes (Leblay & Chekol, 2018). However, these models do not capture temporal dependency and cannot generalize to unobserved time stamps.

**Temporal Modeling on Homogeneous Graphs.** There are attempts on predicting future links on homogeneous graphs (Pareja et al., 2019; Goyal et al., 2018; 2019; Zhou et al., 2018b; Singer et al., 2019). Some of the methods try to incorporate and learn graphical structures to predict future links (Pareja et al., 2019; Zhou et al., 2018b; Singer et al., 2019), while other methods predict by reconstructing an adjacency matrix by using an autoencoder (Goyal et al., 2018; 2019). These

methods seek to predict on single-relational graphs, and are designed to predict future edges in one future step (i.e., for $t + 1$). However, our work focuses on "multi-relational" knowledge graphs and aims for multi-step prediction (i.e., for $t + 1, \ldots, t + k$).

**Recurrent Graph Neural Models.** There have been some studies on recurrent graph neural models for sequential or temporal graph-structured data (Sanchez-Gonzalez et al., 2018; Battaglia et al., 2018; Palm et al., 2018; Seo et al., 2017; Pareja et al., 2019). These methods adopt a message-passing framework for aggregating nodes' neighborhood information (*e.g.*, via graph convolutional operations). GN (Sanchez-Gonzalez et al., 2018; Battaglia et al., 2018) and RRN (Palm et al., 2018) update node representations by a message-passing scheme between time stamps. Some prior methods adopt an RNN to memorize and update the states of node embeddings that are dynamically evolving (Seo et al., 2017), or memorize and update the model parameters for different time stamps (Pareja et al., 2019). In contrast, our proposed method, RE-NET, aims to leverage autoregressive modeling to parameterize the joint probability distributions of events with RNNs.

**Deep Autoregressive Models.** Deep autoregressive models define joint probability distributions as a product of conditionals. DeepGMG (Li et al., 2018) and GraphRNN (You et al., 2018) are deep generative models of graphs and focus on generating homogeneous graphs where there is only a single type of edge. In contrast to these studies, our work focuses on generating heterogeneous graphs, in which multiple types of edges exist, and thus our problem is more challenging. To the best of my knowledge, this is the first paper to formulate the structure inference (prediction) problem for temporal, multi-relational (knowledge) graphs in an autoregressive fashion.

## 3 PROPOSED METHOD: RE-NET

We consider a *temporal knowledge graph* (TKG) as a multi-relational, directed graph with time-stamped edges (relationships) between nodes (entities). An *event* is defined as a time-stamped edge, *i.e.*, (subject entity, relation, object entity, time) and is denoted by a quadruple $(s, r, o, t)$ or $(s_t, r_t, o_t)$. We denote a set of events at time $t$ as $G_t$. A TKG is built upon *a sequence of event quadruples* ordered ascending based on their time stamps, *i.e.*, $\{G_t\}_t = \{(s_i, r_i, o_i, t_i)\}_i$ (with $t_i < t_j, \forall i < j$), where each time-stamped edge has a direction pointing from the subject entity to the object entity.[2] The goal of *learning generative models of events* is to learn a distribution $\mathbb{P}(G)$ over temporal knowledge graphs, based on a set of observed event sets $\{G_1, ..., G_T\}$. To model lasting events which span over a time range, *i.e.*, $(s, r, o, [t_1, t_2])$, we simply partition such event into a sequence of time-stamp events $\{G_{t_1}, ..., G_{t_2}\}$. We leave more sophisticated modeling of lasting events as future work.

### 3.1 RECURRENT EVENT NETWORK

**Sequential Structure Inference in TKG.** The key idea in RE-NET is to define the joint distribution of all the events $G = \{G_1, ..., G_T\}$ in an autoregressive manner, i.e., $\mathbb{P}(G) = \prod_{t=1}^{T} \mathbb{P}(G_t|G_{t-m:t-1})$. Basically, we decompose the joint distribution into a sequence of conditional distributions (e.g., $\mathbb{P}(G_t|G_{t-m:t-1})$), where we assume the probability of the events at a time step, e.g. $G_t$, only depends on the events at the previous $m$ steps, e.g., $G_{t-m:t-1}$. For each conditional distribution $\mathbb{P}(G_t|G_{t-m:t-1})$, we further assume that the events in $G_t$ are mutually independent given the previous events $G_{t-m:t-1}$. In this way, the joint distribution can be rewritten as follows.

$$\mathbb{P}(G) = \prod_t \prod_{(s_t, r_t, o_t) \in G_t} \mathbb{P}(s_t, r_t, o_t|G_{t-m:t-1})$$
$$= \prod_t \prod_{(s_t, r_t, o_t) \in G_t} \mathbb{P}(o_t|s_t, r_t, G_{t-m:t-1}) \cdot \mathbb{P}(r_t|s_t, G_{t-m:t-1}) \cdot \mathbb{P}(s_t|G_{t-m:t-1}). \tag{1}$$

Intuitively, the generation process of each triplet $(s_t, r_t, o_t)$ is defined as below. Given all the past events $G_{t-m:t-1}$, we fist generate a subject entity $s_t$ through the distribution $\mathbb{P}(s_t|G_{t-m:t-1})$. Then we further generate a relation $r_t$ with $\mathbb{P}(r_t|s_t, G_{t-m:t-1})$, and finally the object entity $o_t$ is generated by defining $\mathbb{P}(o_t|s_t, r_t, G_{t-m:t-1})$.

In this work, we assume that $\mathbb{P}(o_t|s_t, r_t, G_{t-m:t-1})$ and $\mathbb{P}(r_t|s_t, G_{t-m:t-1})$ depend only on events that are related to $s$, and focus on modeling the following joint probability:

$$\mathbb{P}(s_t, r_t, o_t|G_{t-m:t-1}) = \mathbb{P}(o_t|s, r, N_{t-m:t-1}^{(s)}) \cdot \mathbb{P}(r_t|s, N_{t-m:t-1}^{(s)}) \cdot \mathbb{P}(s_t|G_{t-m:t-1}), \tag{2}$$

---

[2]The same triple $(s, r, o)$ may occur multiple times in different time stamps, yielding different event quadruples.

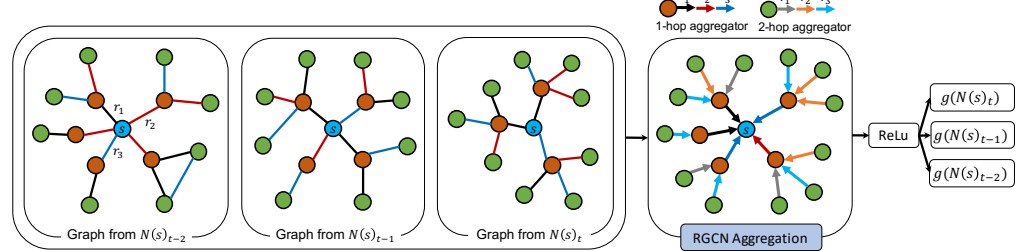

Figure 2: **Illustration of the multi-relational graph (RGCN) aggregator.** The blue node corresponds to node $s$, red nodes are 1-hop neighbors, and green nodes are 2-hop neighbors. Different colored edges are different relations. In this figure, we get $g(N(s)_t), g(N(s)_{t-1})$, and $g(N(s)_{t-2})$ for each graph from a two-layered RGCN aggregator.

where $G_t$ becomes $N_t^{(s)}$ which is a set of neighboring entities interacted with subject entity s under *all* relations at time stamp $t$. For the third probability, the event sets should be considered since subject is not given. Next, we introduce how we parameterize these distributions.

**Recurrent Event Encoder.** RE-NET parameterizes $\mathbb{P}(o_t|s, r, G_{t-m:t-1})$ in the following way:

$$\mathbb{P}(o_t|s, r, N_{t-m:t-1}^{(s)}) \propto \exp\left([\boldsymbol{e}_s : \boldsymbol{e}_r : \boldsymbol{h}_{t-1}(s, r)]^\top \cdot \boldsymbol{w}_{o_t}\right), \tag{3}$$

where $\boldsymbol{e}_s, \boldsymbol{e}_r \in \mathbb{R}^d$ are learnable embedding vectors specified for subject entity s and relation r. $\boldsymbol{h}_{t-1}(s, r) \in \mathbb{R}^d$ is a history vector which encodes the information from the neighbor sets interacted with s in the past, as well as the global information from graph structures of $G_{t-1:t-m}$. Basically, $[\boldsymbol{e}_s : \boldsymbol{e}_r : \boldsymbol{h}_{t-1}(s, r)]$ is an encoding to summarize all the past information. Based on that, we further compute the probability of different object entities $o_t$ by passing the encoding into a linear softmax classifier parameterized by $\{\boldsymbol{w}_{o_t}\}$.

Similarly, we define the probabilities for relations and subjects as follows:

$$\mathbb{P}(r_t|s, N_{t-m:t-1}^{(s)}) \propto \exp\left([\boldsymbol{e}_s : \boldsymbol{h}_{t-1}(s)]^\top \cdot \boldsymbol{w}_{r_t}\right), \tag{4}$$

$$\mathbb{P}(s_t|G_{t-m:t-1}) \propto \exp\left(\boldsymbol{H}_{t-1}^\top \cdot \boldsymbol{w}_{s_t}\right), \tag{5}$$

where $\boldsymbol{h}_{t-1}(s)$ captures all the local information about s in the past, and $\boldsymbol{H}_{t-1} \in \mathbb{R}^d$ is a vector representation to encode the global graph structures $G_{t-1:t-m}$.

For each time step $t$, since the hidden vectors $\boldsymbol{h}_{t-1}(s), \boldsymbol{h}_{t-1}(s, r)$ and $\boldsymbol{H}_{t-1}$ preserve the information from the past events, and we update them in the following recurrent way:

$$\boldsymbol{h}_t(s, r) = \text{RNN}^1(g(N_t^{(s)}), \boldsymbol{H}_t, \boldsymbol{h}_{t-1}(s, r)), \tag{6}$$

$$\boldsymbol{h}_t(s) = \text{RNN}^2(g(N_t^{(s)}), \boldsymbol{H}_t, \boldsymbol{h}_{t-1}(s)), \tag{7}$$

$$\boldsymbol{H}_t = \text{RNN}^3(g(G_t), \boldsymbol{H}_{t-1}), \tag{8}$$

where $g$ is an aggregation function, and $N_t^{(s)}$ stands for all the events related to s at the current time step $t$. Intuitively, we obtain the current information related to s by aggregating all the related events at time $t$, i.e., $g(N_t^{(s)})$. Then we update the hidden vector $\boldsymbol{h}_t(s, r)$ by using the aggregated information $g(N_t^{(s)})$ at the current step, the past value $\boldsymbol{h}_{t-1}(s, r)$ and also the global hidden vector $\boldsymbol{H}_t$. The hidden vector $\boldsymbol{h}_t(s)$ is updated in a similar way. For the aggregation of all events $g(G_t)$, we define $g(G_t) = \max(\{g(N_t^{(s)})\}_s)$, which is from the element-wise max-pooling operation over all $g(N_t^{(s)})$. We use Gated Recurrent Units Cho et al. (2014) as RNN. Details are described in Section A.

For each subject entity s, it can interact with multiple relations and object entities at each time step $t$. In other words, the set $N_t^{(s)}$ can contain multiple events. Designing effective aggregation functions $g$ to aggregate information from $N_t^{(s)}$ for s is therefore a nontrivial problem. Next, we introduce how we design $g(\cdot)$ in RE-NET.

### 3.2 MULTI-RELATIONAL GRAPH (RGCN) AGGREGATOR

Here we discuss the aggregate function $g(\cdot)$, which capture different kinds of neighborhood information for each subject entity and relation, *i.e.*, (s, r). We first introduce two simple aggregation

functions, *i.e.*, mean pooling aggregator and attentive pooling aggregator. These two simple aggregators only collect neighboring entities under the same relation r. Then we introduce a more powerful aggregation function, *i.e.*, multi-relational aggregator.

**Mean Pooling Aggregator.** The baseline aggregator simply takes the element-wise mean of the vectors in $\{e_o : o \in \mathrm{N}_t^{(s,r)}\}$, where $\mathrm{N}_t^{(s,r)}$ is a set of objects interacted with s under r at $t$. But the mean aggregator treats all neighboring objects equally, and thus ignores the different importance of each neighbor entity.

**Attentive Pooling Aggregator.** We define an attentive aggregator based on the additive attention introduced in (Bahdanau et al., 2015) to distinguish the important entities for $(s, r)$. The aggregate function is defined as $g(\mathrm{N}_t^{(s,r)}) = \sum_{o \in \mathrm{N}_t^{(s,r)}} \alpha_o e_o$, where $\alpha_o = \mathrm{softmax}(\mathbf{v}^\top \tanh(\boldsymbol{W}(e_s; e_r; e_o)))$. $\mathbf{v} \in \mathbb{R}^d$ and $\boldsymbol{W} \in \mathbb{R}^{d \times 3d}$ are trainable weight matrices. By adding the attention function of the subject and the relation, the weight can determine how relevant each object entity is to the subject and relation.

**Multi-Relational Aggregator.** Here, we introduce a multi-relational graph aggregator based on (Schlichtkrull et al., 2018). This is a general aggregator that can incorporate information from multi-relational neighbors and multi-hop neighbors. Formally, the aggregator is defined as follows:

$$g(\mathrm{N}_t^{(s)}) = \mathbf{h}_s^{(l+1)} = \sigma\Big(\sum_{r \in R} \sum_{o \in \mathrm{N}_t^{(s,r)}} \frac{1}{c_s} \boldsymbol{W}_r^{(l)} \mathbf{h}_o^{(l)} + \boldsymbol{W}_0^{(l)} \mathbf{h}_s^{(l)}\Big), \tag{9}$$

where initial hidden representations for each node ($\mathbf{h}_o^{(0)}$) are set to trainable embedding vectors ($e_o$) for each node.

Basically, each relation can derive a local graph structure between entities, which further yield a message on each entity by aggregating the information from the neighbors of that entity, i.e., $\sum_{o \in \mathrm{N}_t^{(s,r)}} \frac{1}{c_s} \boldsymbol{W}_r^{(l)} \mathbf{h}_o^{(l)}$. The overall message on each entity is further computed by aggregating all the relation-specific messages, i.e., $\sum_{r \in R} \sum_{o \in \mathrm{N}_t^{(s,r)}} \frac{1}{c_s} \boldsymbol{W}_r^{(l)} \mathbf{h}_o^{(l)}$. Finally, the aggregator $g(\mathrm{N}_t^{(s)})$ is defined by combining both the overall message and the information from past steps, i.e., $\boldsymbol{W}_0^{(l)} \mathbf{h}_s^{(l)}$.

To distinguish between different relations, we introduce independent weight matrices $\{\boldsymbol{W}_r^{(l)}\}$ for each relation $r$. Furthermore, the aggregator collects representations of multi-hop neighbors by introducing multiple layers of the neural network, with each layer indexed by $l$. The number of layers determines the depth to which the node reaches to aggregate information from its local neighborhood. We depict this aggregator in Fig. 2.

The major issue of this aggregator is that the number of parameters grows rapidly with the number of relations. In practice, this can easily lead to overfitting on rare relations and models of very large size. Thus, we adopt the block-diagonal decomposition (Schlichtkrull et al., 2018), where each relation-specific weight matrix is decomposed into a block-diagonal by decomposing into low-dimensional matrices. $\boldsymbol{W}_r^{(l)}$ in equation 9 is defined as a block diagonal matrix, $\mathrm{diag}(\mathbf{A}_{1r}^{(l)}, ..., \mathbf{A}_{Br}^{(l)})$ where $\mathbf{A}_{kr}^{(l)} \in \mathbb{R}^{(d^{(l+1)}/B) \times (d^{(l)}/B)}$ and $B$ is the number of basis matrices. The block decomposition reduces the number of parameters and helps to prevent overfitting.

### 3.3 PARAMETER LEARNING AND INFERENCE OF RE-NET

**Parameter Learning via Event Prediction.** The (object) entity prediction given $(s, r)$ can be viewed as a multi-class classification task, where each class corresponds to one object entity. Similarly, relation prediction given s and subject entity prediction can be considered as a multi-class classification task. Here we omit the notation for previous events. To learn weights and representations for entities and relations, we adopt a multi-class cross-entropy loss to the model's output.The loss function is comprised of three losses and is defined as:

$$\mathcal{L} = - \sum_{(s,r,o,t) \in G} \big(\log(\mathbb{P}(o_t|s_t, r_t) + \lambda_1 \log(\mathbb{P}(r_t|s_t)) + \lambda_2 \log(\mathbb{P}(s_t))), \tag{10}$$

where $G$ is set of events, and $\lambda_1$ and $\lambda_2$ are importance parameters that control the importance of each loss term. $\lambda_1$ and $\lambda_2$ can chosen depending on a task. If the task aims to predict $o$ given $(s, r)$, then we can give small values to $\lambda_1$ and $\lambda_2$. Each probability is defined in equations 3, 4, and 5, respectively. We apply teacher forcing for model training over historical data.

---

**Algorithm 1:** Inference algorithm of RE-NET

---

**Input:** Observed graph sequence: $\{G_1, ..., G_{t-1}\}$, Number of events to sample at each step: $M$.
**Output:** An estimation of the conditional distribution: $\mathbb{P}(G_{t+\Delta t}|G_{:t})$.

1   $t' = t$
2   **while** $t' \leq t + \Delta t$ **do**
3      Sample $M$ number of s $\sim \mathbb{P}(s|\hat{G}_{t+1:t'-1}, G_{:t})$ by Equation 5.
4      Pick top-$k$ triples $\{(s_1, r_1, o_1, t'), ..., (s_k, r_k, o_k, t')\}$ ranked by Equation 2.
5      $\hat{G}_{t'} = \{(s_1, r_1, o_1, t'), ..., (s_k, r_k, o_k, t')\}$
6      $t' = t' + 1$
7   Estimate the probability of each event $\mathbb{P}(s, r, o|\hat{G}_{t+1:t+\Delta t-1}, G_{:t})$ by Equation 2.
8   Estimate the joint distribution of all events $\mathbb{P}(G_{t+\Delta t}|\hat{G}_{t+1:t+\Delta t-1}, G_{:t})$ by Equation 1.
9   **return** $\mathbb{P}(G_{t+\Delta t}|\hat{G}_{t+1:t+\Delta t-1}, G_{:t})$ as the estimation.

---

**Multi-step Inference over Time.** At inference time, RE-NET seeks to predict the forthcoming events based on the previous observations. Suppose that the current time is $t$ and we aim at predicting events at time $t + \Delta t$, then the problem of multi-step inference can be formalized as an inference problem, i.e., inferring the conditional probability $\mathbb{P}(G_{t+\Delta t}|G_{:t})$. The problem is nontrivial as we need to integrate over all $G_{t+1:t+\Delta t-1}$. To achieve efficient inference, we draw a sample of $G_{t+1:t+\Delta t-1}$, and estimate the conditional probability in the following way:

$$\mathbb{P}(G_{t+\Delta t}|G_{:t}) = \sum_{G_{t+1:t+\Delta t-1}} \mathbb{P}(G_{t+\Delta t}|G_{:t+\Delta t-1})\mathbb{P}(G_{t+\Delta t-1}|G_{:t+\Delta t-2}) \cdots \mathbb{P}(G_{t+1}|G_{:t})$$

$$= \mathbb{E}_{\mathbb{P}(G_{t+1:t+\Delta t-1}|G_{:t})}[\mathbb{P}(G_{t+\Delta t}|G_{:t+\Delta t-1})] \simeq \mathbb{P}(G_{t+\Delta t}|\hat{G}_{t+1:t+\Delta t-1}, G_{:t}) \quad (11)$$

Such an inference procedure is intuitive. Basically, one starts with computing $\mathbb{P}(G_{t+1}|G_{:t})$, and drawing a sample $\hat{G}_{t+1}$ from the conditional distribution. With this sample, one can further compute $\mathbb{P}(G_{t+2}|\hat{G}_{t+1}, G_{:t})$. By iteratively computing the conditional distribution for $G_{t'}$ and drawing a sample from it, one can eventually estimate $\mathbb{P}(G_{t+\Delta t}|G_{:t})$ as $\mathbb{P}(G_{t+\Delta t}|\hat{G}_{t+1:t+\Delta t-1}, G_{:t})$. In practice, we can improve the estimation by drawing multiple graph samples at each step, but RE-NET already performs very well with a single sample, and thus we only draw one sample graph at each step for better efficiency. Based on the estimation of the conditional distribution, we can further predict events which are likely to form in the future. We summarize the detailed inference algorithm in Algorithm 1. In Algorithm 1, we sample one graph at a time. To obtain the graph, we first sample $M$ number of s (line 3) and pick top-$k$ triples (line 4). Then we have a knowledge graph at time $t'$ (line 5).

**Computational Complexity Analysis.** Here we analyze the time complexity of the graph generation algorithm 1. To compute $\mathbb{P}(s_t|G_{t-m:t-1})$ (equation 5), it takes $O(|E|Lm)$, where $|E|$ is the maximum number of triples among $\{G_{t-m}, ..., G_{t-1}\}$, $L$ is the number of layers of aggregation, and $m$ is the number of the past time steps since we unroll $m$ time steps in RNN. From this probability, we sample $M$ number of subjects $s$. To compute $\mathbb{P}(s_t, r_t, o_t|G_{t-m:t-1})$ (equation 2), it takes $O(D^L m)$, where $D$ is the maximum degree of entities. To get probabilities of all possible triples given sampled subjects, it needs $O(M|R||O|D^L m)$ where $|R|$ is the total number of relations and $|O|$ is the total number of entities. Thus, the time complexity for generating one graph is $O(|E|Lm + M|R||O|(D^L m + \log k))$ where $k$ is the cutoff number for picking top-$k$ triples. The time complexity is linear to the number of entities and relations, and the number of sampled $s$.

## 4   EXPERIMENTS

Evaluating the quality of generated graphs is challenging, especially in knowledge graphs (Theis et al., 2015). Instead, we evaluate our proposed method on a link prediction task on temporal knowledge graphs. The task of predicting future links aims to predict unseen relationships with object entities given $(s, r, ?, t)$ (or subject entities given $(?, r, o, t)$), based on the observed events in the TKG. Essentially, the task is a ranking problem over all the events $(s, r, ?, t)$ (or $(?, r, o, t)$). RE-NET can approach this problem by computing the probability of each event in a distant future with the inference algorithm in Algorithm 1, and further rank all the events according to their probabilities.

We evaluate our proposed method on three benchmark tasks: (1) predicting future events on three event-based datasets; (2) predicting future facts on two knowledge graphs which include facts with time spans, and (3) studying parameter sensitivity and ablation of our proposed method. Section 4.1

Table 1: Performance comparison on temporal link prediction (average metrics in % over 5 runs) on three event-based TKG datasets with filtered setting. RE-NET achieves the best results. Results with raw setting are in the supplementary material.

| | Method | ICEWS18 - filtered | | | | GDELT - filtered | | | | ICEWS14 - filtered | | | |
|---|---|---|---|---|---|---|---|---|---|---|---|---|---|
| | | MRR | H@1 | H@3 | H@10 | MRR | H@1 | H@3 | H@10 | MRR | H@1 | H@3 | H@10 |
| Static | TransE | 17.56 | 2.48 | 26.95 | 43.87 | 16.05 | 0.00 | 26.10 | 42.29 | 18.65 | 1.21 | 31.34 | 47.07 |
| | DistMult | 22.16 | 12.13 | 26.00 | 42.18 | 18.71 | 11.59 | 20.05 | 32.55 | 19.06 | 10.09 | 22.00 | 36.41 |
| | ComplEx | 30.09 | 21.88 | 34.15 | 45.96 | 22.77 | 15.77 | 24.05 | 36.33 | 24.47 | 16.13 | 27.49 | 41.09 |
| | R-GCN | 23.19 | 16.36 | 25.34 | 36.48 | 23.31 | 17.24 | 24.94 | 34.36 | 26.31 | 18.23 | 30.43 | 45.34 |
| | ConvE | 36.67 | 28.51 | 39.80 | 50.69 | 35.99 | 27.05 | 39.32 | 49.44 | 40.73 | 33.20 | 43.92 | 54.35 |
| | RotatE | 23.10 | 14.33 | 27.61 | 38.72 | 22.33 | 16.68 | 23.89 | 32.29 | 29.56 | 22.14 | 32.92 | 42.68 |
| Temporal | HyTE | 7.31 | 3.10 | 7.50 | 14.95 | 6.37 | 0.00 | 6.72 | 18.63 | 11.48 | 5.64 | 13.04 | 22.51 |
| | TTransE | 8.36 | 1.94 | 8.71 | 21.93 | 5.52 | 0.47 | 5.01 | 15.27 | 6.35 | 1.23 | 5.80 | 16.65 |
| | TA-DistMult | 28.53 | 20.30 | 31.57 | 44.96 | 29.35 | 22.11 | 31.56 | 41.39 | 20.78 | 13.43 | 22.80 | 35.26 |
| | Know-Evolve* | 3.27 | 3.23 | 3.23 | 3.26 | 2.43 | 2.33 | 2.35 | 2.41 | 1.42 | 1.35 | 1.37 | 1.43 |
| | Know-Evolve+MLP | 9.29 | 5.11 | 9.62 | 17.18 | 22.78 | 15.40 | 25.49 | 35.41 | 22.89 | 14.31 | 26.68 | 38.57 |
| | DyRep+MLP | 9.86 | 5.14 | 10.66 | 18.66 | 23.94 | 15.57 | 27.88 | 36.58 | 24.61 | 15.88 | 28.87 | 39.34 |
| | R-GCRN+MLP | 35.12 | 27.19 | 38.26 | 50.49 | 37.29 | 29.00 | 41.08 | 51.88 | 36.77 | 28.63 | 40.15 | 52.33 |
| | RE-NET w/o multi-step | 40.05 | 33.32 | 42.60 | 52.92 | 38.72 | 30.57 | 42.52 | 52.78 | 42.72 | 35.42 | 46.06 | 56.15 |
| | RE-NET w/o agg. | 33.46 | 26.64 | 35.98 | 46.62 | 38.10 | 29.34 | 41.26 | 51.61 | 42.23 | 34.73 | 45.61 | 56.07 |
| | RE-NET w. mean agg. | 40.70 | 34.24 | 43.27 | 53.65 | 38.35 | 29.92 | 42.13 | 52.52 | 43.79 | 36.21 | 47.34 | 57.47 |
| | RE-NET w. attn agg. | 40.96 | 34.57 | 44.08 | 54.32 | 38.54 | 29.65 | 42.25 | 52.85 | 43.94 | 37.01 | 47.85 | 57.91 |
| | RE-NET | **42.93** | **36.19** | **45.47** | **55.80** | **40.12** | **32.43** | **43.40** | **53.80** | **45.71** | **38.42** | **49.06** | **59.12** |
| | RE-NET w. GT $(s, r)$ | 44.33 | 37.61 | 46.83 | 57.27 | 41.80 | 33.54 | 45.71 | 56.03 | 46.74 | 39.41 | 50.10 | 60.19 |

summarizes the datasets, and the supplementary material contains additional information. In all these experiments, we perform predictions on time stamps that are not observed during training.

## 4.1 Experimental Set-up

**Datasets.** We use five datasets: 1) three event-based temporal knowledge graphs: ICEWS18 (Boschee et al., 2015), ICEWS14 (Trivedi et al., 2017), and GDELT (Leetaru & Schrodt, 2013); and 2) two knowledge graphs where temporally associated facts have meta-facts as $(s, r, o, [t_s, t_e])$ where $t_s$ is the starting time point and $t_e$ is the ending time point: WIKI (Leblay & Chekol, 2018) and YAGO (Mahdisoltani et al., 2014). The details of the datasets are described in Section B.

**Evaluation Setting and Metrics.** For each dataset except ICEWS14, we split it into three subsets, i.e., train(80%)/valid(10%)/test(10%), by time stamps. Thus, (times of train) < (times of valid) < (times of test). We report Mean Reciprocal Ranks (MRR) and Hits@1/3/10, using the filtered version and the raw version of the datasets. Similar to the definition of filtered setting in (Bordes et al., 2013), during evaluation, we remove from the list of corrupted triplets all the triplets that appear either in the train, dev, or test set.

**Competitors.** We compare our approach to baselines for static graphs and temporal graphs:

**(1)** *Static Methods.* By ignoring the edge time stamps, we construct a static, cumulative graph for all the training events, and apply multi-relational graph representation learning methods including TransE (Bordes et al., 2013), DistMult (Yang et al., 2015), ComplEx (Trouillon et al., 2016), R-GCN (Schlichtkrull et al., 2018), ConvE (Dettmers et al., 2018), and RotatE (Sun et al., 2019).

**(2)** *Temporal Reasoning Methods.* We also compare state-of-the-art temporal reasoning methods for knowledge graphs, including Know-Evolve[3] (Trivedi et al., 2017), TA-DistMult (García-Durán et al., 2018), HyTE (Dasgupta et al., 2018), and TTransE (Leblay & Chekol, 2018). TA-DistMult, HyTE, and TTransE are for a interpolation task which is to make predictions at time $t$ such that $t_1 < t < t_2$, which is different from our setting. We give random values or embeddings that are not observed during training. To see the effectiveness of our recurrent event encoder, we use encoders of previous work and our MLP decoder as baselines; we compare Know-Evolve, Dyrep (Trivedi et al., 2019), and GCRN (Seo et al., 2017) combined with our MLP decoder, which are called Know-Evolve+MLP, DyRep+MLP, and R-GCRN+MLP. The GCRN utilizes Graph Gonvolutional Network (Kipf & Welling, 2016). Instead, we use RGCN (Schlichtkrull et al., 2018) to deal with multi-relational graphs.

---

[3]*: We found a problematic formulation in Know-Evolve when dealing with concurrent events (Eq. (3) in its paper) and a flaw in its evaluation code. The performance dramatically drops after fixing the evaluation code. Details of this issues are discussed in Section F.

Table 2: Performance comparison on temporal link prediction (average metrics in % over 5 runs) on two public temporal knowledge graphs, i.e., WIKI and YAGO.

| | Method | WIKI - filtered | | | WIKI - raw | | | YAGO - filtered | | | YAGO - raw | | |
|---|---|---|---|---|---|---|---|---|---|---|---|---|---|
| | | MRR | H@3 | H@10 | MRR | H@3 | H@10 | MRR | H@3 | H@10 | MRR | H@3 | H@10 |
| Static | TransE | 46.68 | 49.71 | 51.71 | 26.21 | 31.25 | 39.06 | 48.97 | 62.45 | 66.05 | 33.85 | 48.19 | 59.50 |
| | DistMult | 46.12 | 49.81 | 51.38 | 27.96 | 32.45 | 39.51 | 59.47 | 60.91 | 65.26 | 44.05 | 49.70 | 59.94 |
| | ComplEx | 47.84 | 50.08 | 51.39 | 27.69 | 31.99 | 38.61 | 61.29 | 62.28 | 66.82 | 44.09 | 49.57 | 59.64 |
| | R-GCN | 37.57 | 39.66 | 41.90 | 13.96 | 15.75 | 22.05 | 41.30 | 44.44 | 52.68 | 20.25 | 24.01 | 37.30 |
| | ConvE | 47.57 | 50.10 | 50.53 | 26.03 | 30.51 | 39.18 | 62.32 | 63.97 | 65.60 | 41.22 | 47.03 | 59.90 |
| | RotatE | 50.67 | 50.74 | 50.88 | 26.08 | 31.63 | 38.51 | 65.09 | 65.67 | 66.16 | 42.08 | 46.77 | 59.39 |
| Temporal | HyTE | 43.02 | 45.12 | 49.49 | 25.40 | 29.16 | 37.54 | 23.16 | 45.74 | 51.94 | 14.42 | 39.73 | 46.98 |
| | TTransE | 31.74 | 36.25 | 43.45 | 20.66 | 23.88 | 33.04 | 32.57 | 43.39 | 53.37 | 26.10 | 36.28 | 47.73 |
| | TA-DistMult | 48.09 | 49.51 | 51.70 | 26.44 | 31.36 | 38.97 | 61.72 | 65.32 | 67.19 | 44.98 | 50.64 | 61.11 |
| | Know-Evolve* | 0.09 | 00.03 | 0.10 | 0.03 | 0 | 0.04 | 00.07 | 0 | 0.04 | 0.02 | 0 | 0.01 |
| | Know-Evolve+MLP | 12.64 | 14.33 | 21.57 | 10.54 | 13.08 | 20.21 | 6.19 | 6.59 | 11.48 | 5.23 | 5.63 | 10.23 |
| | DyRep+MLP | 11.60 | 12.74 | 21.65 | 10.41 | 12.06 | 20.93 | 5.87 | 6.54 | 11.98 | 4.98 | 5.54 | 10.19 |
| | R-GCRN+MLP | 47.71 | 48.14 | 49.66 | 28.68 | 31.44 | 38.58 | 53.89 | 56.06 | 61.19 | 43.71 | 48.53 | 56.98 |
| | RE-NET w/o multi-step | 51.01 | 51.14 | 52.91 | 29.91 | 32.60 | 40.29 | 64.21 | 64.70 | 67.11 | 45.88 | 51.78 | 60.97 |
| | RE-NET w/o agg. | 31.08 | 33.98 | 45.53 | 17.55 | 20.65 | 33.51 | 33.86 | 36.89 | 50.72 | 27.37 | 30.20 | 46.35 |
| | RE-NET w. mean agg. | 51.13 | 51.37 | 53.01 | 30.19 | 32.94 | 40.57 | 65.10 | 65.24 | 67.34 | 46.33 | 52.49 | 61.21 |
| | RE-NET w. attn agg. | 51.25 | 52.54 | 53.12 | 30.25 | 30.12 | 40.86 | 65.13 | 67.54 | 67.87 | 46.56 | 52.56 | 61.35 |
| | RE-NET | **51.97** | **52.07** | **53.91** | **30.87** | **33.55** | **41.27** | **65.16** | **65.63** | **68.08** | **46.81** | **52.71** | **61.93** |
| | RE-NET w. GT $(s, r)$ | 53.57 | 54.10 | 55.72 | 32.44 | 35.42 | 43.16 | 66.80 | 67.23 | 69.77 | 48.60 | 54.20 | 63.59 |

**(3)** *Variants of* **RE-NET.** To evaluate the importance of different components of RE-NET, we varied our base model in different ways: RE-NET w/o multi-step which does not update history during inference, RE-NET without the aggregator (RE-NET w/o agg.), RE-NET with a mean aggregator (RE-NET w. mean agg.), and RE-NET with an attentive aggregator (RE-NET w. attn agg.). takes a zero vector instead of a representation of the aggregator. RE-NET w. GT $(s, r)$ denotes RE-NET with ground truth history or interactions during multi-step inference, and thus the model knows all the interactions before the time for testing. It does not update history (or generate a graph) since it already has ground truth history. Experiment settings and implementation details of RE-NET and baselines are described in Section C.

## 4.2    PERFORMANCE COMPARISON ON TEMPORAL KNOWLEDGE GRAPHS.

In this section we compare our proposed method with other baselines. The test results are obtained by averaged metrics over the entire test sets on datasets.

**Performances on Event-based TKGs.** Table 1 summarizes results on three event-based datasets: ICEWS18, GDELT, and ICEWS14. Our proposed RE-NET outperforms all other baselines on these datasets. Static methods show good results but they underperform our method since they do not consider temporal factors. Also, RE-NET outperforms all other temporal methods, which demonstrates effectiveness of the proposed method. The modified Know-Evolve with our MLP decoder (Know-Evovle+MLP) achieves the better performances than Know-Evolve, which shows effectiveness of our MLP decoder, but there is still a large gap from our model. We notice that Know-Evolve and DyRep has a gradient exploding issue on their encoder since their RNN-like structures keep accumulating embedding over time. This issue degrades their performances. Graph Convolutional Recurrent Network (GCRN) is not for dynamic and multi-relational graphs, and is not capable of link prediction. We modified the model to work on dynamic graphs and our problem setting by using RGCN instead of GCN, and our MLP decoder. The modified model (R-GCRN+MLP) shows good performances but it does not outperform our method. R-GCRN+MLP has a similar structure to ours in that it has a recurrent encoder and an RGCN aggregator but it lacks multi-step inference, global information, and sophisticated modeling for the recurrent encoder. These results of the combined models suggest the our recurrent event encoder shows better performances in link prediction. Importantly, all these temporal methods are not capable of multi-step inference, while RE-NET sequentially infers multi-step events.

**Performances on Public KGs.** Previous results have proved the effectiveness of RE-NET, and here we will compare the method on the Public KGs: WIKI and YAGO. In Table 2, our proposed RE-NET outperforms all other baselines. In these datasets, baselines show better results than in the Event-based TKGs. This is due to the characteristics of the datasets; they have facts that are valid within a time span. However, our proposed method consistently outperforms the static and temporal

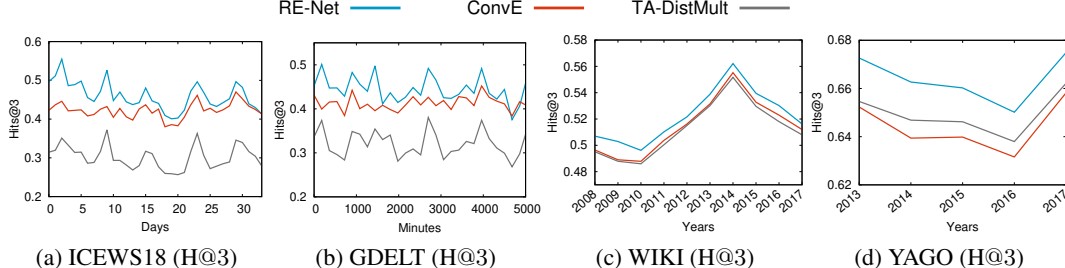

Figure 3: Performance of temporal link prediction over future time stamps with filtered Hits@3. RE-NET consistently outperforms the baselines.

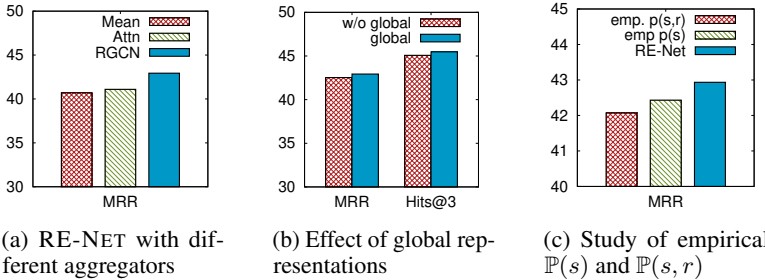

(a) RE-NET with different aggregators

(b) Effect of global representations

(c) Study of empirical $\mathbb{P}(s)$ and $\mathbb{P}(s,r)$

Figure 4: Performance study on model variations. We study the effects of (a) RE-NET with different aggregators, (b) effect of the global representation from a global graph structure, and (c) empirical $\mathbb{P}(\mathrm{s})$ and $\mathbb{P}(\mathrm{s},\mathrm{r})$

methods. which implies that RE-NET effectively infers new events using a powerful event encoder and an aggregator, and provides accurate prediction results.

**Performances of Prediction over Time.** Next, we further study performances of RE-NET over time. Figs. 3 shows the performance comparisons over different time stamps on the ICEWS18, GDELT, WIKI, and YAGO datasets with filtered Hits@3 metrics. RE-NET consistently outperforms baseline methods for all different time stamps. Performances of each method fluctuate since testing entities are different at each time step. We notice that with the increase of time step, the difference between RE-NET and ConvE is getting smaller as shown in Fig. 3. This is expected since further future events are harder to predict. Furthermore, we can think that the decline of the performances is due to the generation of a long graph sequence. To estimate the joint probability distribution of all events in a distant future, RE-NET should generates a long sequence of graphs. The quality of generated graphs deteriorates when RE-NET generates a long graph sequence.

### 4.3 ABLATION STUDY

In this section, we study the effect of variations in RE-NET. To evaluate the importance of different components of RE-NET, we varied our base model in different ways, measuring the change in performance on the link prediction task on the ICEWS18 dataset. We present the results in Tables 1, 2, and Figs. 4.

**Different Aggregators.** We first analyze the effect of the aggregator. In Tables 1, 2, we observe that RE-NET w/o agg. hurts model quality. This suggests that introducing aggregators make the model capable of dealing with concurrent events and aggregators improve the prediction performances. Fig. 4a shows the performances of RE-NET with different aggregators. Among them, RGCN aggregator outperforms other aggregators. This aggregator has the advantage of exploring multi-relational neighbors not limited to neighbors under the same relation. Also, RE-NET with an attentive aggregator shows better performances than RE-NET with a mean aggregator, which implies that giving different attention weights to each neighbor helps predictions.

**Global Information.** We further observe that representations from global graph structures help the predictions. Fig. 4b shows effectiveness of a representation of global graph structures. We consider that global representations give information beyond local graph structures.

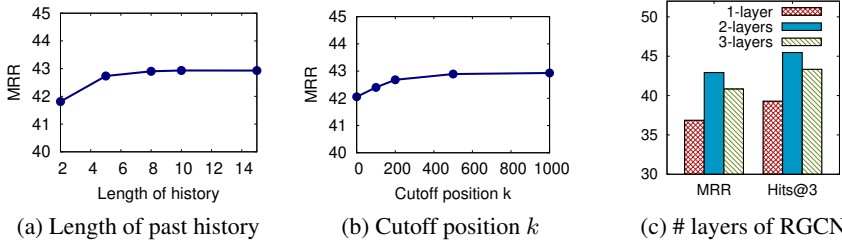

(a) Length of past history     (b) Cutoff position $k$     (c) # layers of RGCN

Figure 5: Parameter sensitivity on RE-NET. We study the effects of (a) length of RNN history in event sequence encoder, (b) cutoff position at inference time, and (c) number of RGCN layers in neigborhood aggregation.

**Empirical Probabilities.** Here, we study the role of $\mathbb{P}(s_t|G_{t-m:t-1})$ and $\mathbb{P}(r_t|s, G_{t-m:t-1})$. We simply denote them as $\mathbb{P}(s)$ and $\mathbb{P}(r)$ for brevity. Also, $\mathbb{P}(s_t, r_t|G_{t-m:t-1})$ (or simply $\mathbb{P}(s, r)$) is equivalent to $\mathbb{P}(s)\mathbb{P}(r)$. In Fig 4c, emp. $\mathbb{P}(s)$ denotes a model with empirical $\mathbb{P}(s)$ (or $\mathbb{P}_e(s)$) which is defined as $\mathbb{P}_e(s) =$ (# of s-related triples) / (total # of triples). Also, emp. $\mathbb{P}(s, r)$ denotes a model with $\mathbb{P}_e(s)$ and $\mathbb{P}_e(r)$ which is defined as $\mathbb{P}_e(r) =$ (# of r-related triples) / (total # of triples). Thus, $\mathbb{P}_e(s, r) = \mathbb{P}_e(s)\mathbb{P}_e(r)$. RE-NET use a trained $\mathbb{P}(s)$ and $\mathbb{P}(r)$. The results show that the trained $\mathbb{P}(s)$ and $\mathbb{P}(r)$ help RE-NET for multi-step predictions. Using $\mathbb{P}_s(s)$ underperforms RE-NET, and using $\mathbb{P}_e(s, r) = \mathbb{P}_e(s)\mathbb{P}_e(r)$ shows the worst performances, which suggests that training each part of the probability in equation 2 gives better prediction performances.

## 4.4 SENSITIVITY ANALYSIS

In this section, we study the parameter sensitivity of RE-NET including the length of history for the event encoder, cutoff position k for events to generate a graph. Furthermore, we study the layers of RGCN aggregator. We report the performance change of RE-NET on the ICEWS18 dataset by varying the hyper-parameters in Table 5.

**Length of Past History in Recurrent Event Encoder.** The recurrent event encoder takes the sequence of past interactions up to $m$ graph sequences or previous histories. Figure 5a shows the performances with varying length of past histories. When RE-NET uses longer histories, MRR is getting higher. However, the MRR is not likely to go higher when the length of history is 5 and over. This implies that long history does not make big differences.

**Cut-off Position $k$ at Inference Time.** To generate a graph at each time, we cut off top-$k$ triples on ranking results. Fig. 5b shows the performances with choosing different cutoff position $k$. When $k$ is 0, RE-NET does not generate graphs for estimating $\mathbb{P}(G_{t+\Delta t}|G_{:t})$, and it shows the lowest result. which means RE-NET performs single-step predictions, . When $k$ is larger, the performance is getting higher and it is saturated after 500. We notice that the conditional distribution $\mathbb{P}(G_{t+\Delta t}|G_{:t})$ can be approximated by $\mathbb{P}(G_{t+\Delta t}|\hat{G}_{t+1:t+\Delta t-1}, G_{:t})$ by using a larger cutoff position.

**Layers of RGCN Aggregator.** We examine the number of layers in the RGCN aggregator. The number of layers in the aggregator means the depth to which the node reaches. Fig. 5c shows the performances according to different numbers of layers of RGCN. We notice that 2-layered RGCN improves the performances considerably compared to 1-layered RGCN since 2-layered RGCN aggregates more information. However, RE-NET with 3-layered RGCN underperforms RE-NET with 2-layered RGCN. We conjecture that the bigger parameter space leads to overfitting.

## 5 CONCLUSION

In this work, we studied the sequential graph generation on temporal knowledge graphs. To tackle this problem, we proposed Recurrent Event Network (RE-NET) which models temporal, multi-relational, and concurrent interactions between entities. A recurrent event encoder in RE-NET summarizes information of the past event sequences, and a neighborhood aggregator collects the information of concurrent. RE-NET defines the joint probability of all events, and thus is capable of inferring global structures in a sequential manner. We tested the proposed model on a link prediction task on temporal knowledge graphs. The experiment revealed that the proposed RE-NET outperforms all the static and temporal methods and our extensive experiments shows its strength. Interesting future work includes modeling lasting events and performing inference on the long-lasting graph structures.

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

## A    RECURRENT EVENT ENCODER

We define a recurrent event encoder based on RNN as follows:

$$\boldsymbol{h}_t(\text{s}, \text{r}) = \text{RNN}(g(\text{N}_t(\text{s})), \boldsymbol{H}_t, \boldsymbol{h}_{t-1}(\text{s}, \text{r})).$$

We use Gated Recurrent Units (Cho et al., 2014) as RNN:

$$\boldsymbol{a}_t = [\boldsymbol{e}_\text{s} : \boldsymbol{e}_\text{r} : g(\text{N}_t(\text{s})) : \boldsymbol{H}_t]$$
$$\boldsymbol{z}_t = \sigma(\boldsymbol{W}_z \boldsymbol{a}_t + \boldsymbol{U}_z \boldsymbol{h}_{t-1})$$
$$\boldsymbol{r}_t = \sigma(\boldsymbol{W}_r \boldsymbol{a}_t + \boldsymbol{U}_r \boldsymbol{h}_{t-1})$$
$$\boldsymbol{h}_t = (1 - \boldsymbol{z}_t) \circ \boldsymbol{h}_{t-1} + \boldsymbol{z}_t \circ \tanh(\boldsymbol{W}_h \boldsymbol{a}_t + \boldsymbol{U}_h(\boldsymbol{r}_t \circ \boldsymbol{h}_{t-1})),$$

where : is concatenation, $\sigma(\cdot)$ is an activation function, and $\circ$ is a Hadamard operator. The input is a concatenation of four vectors: subject embedding, object embedding, aggregation of neighborhood representations, and global information vector $(\boldsymbol{e}_\text{s}, \boldsymbol{e}_\text{r}, g(\text{N}_t(\text{s})), \boldsymbol{H}_t)$. $\boldsymbol{h}_t(s)$ and $\boldsymbol{H}_t$ are similarly defined. For $\boldsymbol{h}_t(s)$, a concatenation of subject embedding, aggregation of neighborhood representations, and global information vector $(\boldsymbol{e}_\text{s}, g(\text{N}_t(s)), \boldsymbol{H}_t)$ is input. For $\boldsymbol{H}_t$, aggregation of the whole graph representations $g(G_t)$ is input.

## B    DATASET

We use five datasets: 1) three event-based temporal knowledge graphs and 2) two knowledge graphs where temporally associated facts have meta-facts as $(\text{s}, \text{r}, \text{o}, [t_s, t_e])$ where $t_s$ is the starting time point and $t_e$ is the ending time point. The first group of graphs includes Integrated Crisis Early Warning System (ICEWS18  (Boschee et al., 2015) and ICEWS14 (Trivedi et al., 2017)), and Global Database of Events, Language, and Tone (GDELT) (Leetaru & Schrodt, 2013). The second group of graphs includes WIKI (Leblay & Chekol, 2018) and YAGO (Mahdisoltani et al., 2014).

ICEWS18 is collected from 1/1/2018 to 10/31/2018, ICEWS14 is from 1/1/2014 to 12/31/2014, and GDELT is from 1/1/2018 to 1/31/2018. The ICEWS14 is from (Trivedi et al., 2017). We didn't use their version of the GDELT dataset since they didn't release the dataset.

WIKI and YAGO datasets have temporally associated facts $(\text{s}, \text{r}, \text{o}, [t_s, t_e])$. We preprocess the datasets such that each fact is converted to $\{(\text{s}, \text{r}, \text{o}, t_s), (\text{s}, \text{r}, \text{o}, t_\text{s} + 1_t), ..., (\text{s}, \text{r}, \text{o}, t_e)\}$ where $1_t$ is a unit time to ensure each fact has a sequence of events. Noisy events of early years are removed (before 1786 for WIKI and 1830 for YAGO).

The difference between the first group and the second group is that facts happen multiple times (even periodically) on the first group (event-based knowledge graphs) while facts last long time but are not likely to occur multiple times in the second group.

Dataset statistics are described on table 3.

Table 3: Dataset Statistics.

| Data | $N_{train}$ | $N_{valid}$ | $N_{test}$ | $N_{ent}$ | $N_{rel}$ | Time granularity |
|---|---|---|---|---|---|---|
| GDELT | 1,734,399 | 238,765 | 305,241 | 7,691 | 240 | 15 mins |
| ICEWS18 | 373,018 | 45,995 | 49,545 | 23,033 | 256 | 24 hours |
| ICEWS14 | 323,895 | - | 341,409 | 12,498 | 260 | 24 hours |
| WIKI | 539,286 | 67,538 | 63,110 | 12,554 | 24 | 1 year |
| YAGO | 161,540 | 19,523 | 20,026 | 10,623 | 10 | 1 year |

## C    DETAILED EXPERIMENTAL SETTINGS

**Model details of RE-NET.** We use Gated Recurrent Units (Cho et al., 2014) as our recurrent event encoder, where the length of history is set as $m = 10$ which means saving past 10 event sequences. If the events related to s are sparse, we check the previous time steps until we get $m$ previous time steps related to the entity s. We pretrain the parameters related to equations 5 and 8 due to large size of training graphs. We use a multi-relational aggregator to compute $\boldsymbol{H}_t$. The aggregator provides hidden representations for each node and we max-pool over all hidden representations to get $\boldsymbol{H}_t$. At inference time, RE-NET performs multi-step prediction across the time stamps in dev and test sets. In each time step, we sample 1000 $(= M)$ number of subjects and save top-1000 $(= k)$ triples to use

Table 4: Performance comparison on ICEWS and GDELT datasets with raw metrics. We observe our method outperforms all other methods.

|  | Method | ICEWS18 - raw | | | | GDELT - raw | | | | ICEWS14 - raw | | | |
|---|---|---|---|---|---|---|---|---|---|---|---|---|---|
|  |  | MRR | H@1 | H@3 | H@10 | MRR | H@1 | H@3 | H@10 | MRR | H@1 | H@3 | H@10 |
| Static | TransE | 12.37 | 1.51 | 15.99 | 34.65 | 7.84 | 0.00 | 8.92 | 23.30 | 11.17 | 0.73 | 14.45 | 32.29 |
|  | DisMult | 13.86 | 5.61 | 15.22 | 31.26 | 8.61 | 3.91 | 8.27 | 17.04 | 9.72 | 3.23 | 10.09 | 22.53 |
|  | ComplEx | 15.45 | 8.04 | 17.19 | 30.73 | 9.84 | 5.17 | 9.58 | 18.23 | 11.20 | 5.68 | 12.11 | 24.17 |
|  | R-GCN | 15.05 | 8.13 | 16.49 | 29.00 | 12.17 | 7.40 | 12.37 | 20.63 | 15.03 | 7.17 | 16.12 | 31.47 |
|  | ConvE | 22.81 | 13.63 | 25.83 | 41.43 | 18.37 | 11.29 | 19.36 | 32.13 | 21.32 | 12.83 | 23.45 | 38.44 |
|  | RotatE | 11.63 | 4.21 | 12.31 | 28.03 | 3.62 | 0.52 | 2.26 | 8.37 | 9.79 | 3.77 | 9.37 | 22.24 |
| Temporal | HyTE | 7.41 | 3.10 | 7.33 | 16.01 | 6.69 | 0.01 | 7.57 | 19.06 | 7.72 | 1.65 | 7.94 | 20.16 |
|  | TTransE | 8.44 | 1.85 | 8.95 | 22.38 | 5.53 | 0.46 | 4.97 | 15.37 | 4.34 | 0.81 | 3.27 | 10.47 |
|  | TA-DistMult | 15.62 | 7.63 | 17.09 | 32.21 | 10.34 | 4.44 | 10.44 | 21.63 | 11.29 | 5.11 | 11.60 | 23.71 |
|  | Know-Evolve* | 0.11 | 0.00 | 0.00 | 0.47 | 0.11 | 0.00 | 0.02 | 0.10 | 0.05 | 0.00 | 0.00 | 0.10 |
|  | Know-Evolve+MLP | 7.41 | 3.31 | 7.87 | 14.76 | 15.88 | 11.66 | 15.69 | 22.28 | 16.81 | 9.95 | 18.63 | 29.20 |
|  | DyRep+MLP | 7.82 | 3.57 | 7.73 | 16.33 | 16.25 | 11.78 | 16.45 | 23.86 | 17.54 | 10.39 | 19.87 | 30.34 |
|  | R-GCRN+MLP | 23.46 | 14.24 | 26.62 | 41.96 | 18.63 | 11.53 | 19.80 | 32.42 | 21.39 | 12.74 | 23.60 | 38.96 |
|  | RE-NET w/o multi-step | 25.67 | 15.98 | 29.33 | 44.65 | 19.15 | 11.87 | 20.34 | 33.39 | 22.55 | 13.46 | 25.36 | 41.45 |
|  | RE-NET w/o agg. | 23.11 | 14.46 | 26.45 | 39.96 | 18.90 | 11.69 | 20.07 | 32.93 | 21.43 | 12.25 | 24.12 | 40.09 |
|  | RE-NET w. mean agg. | 25.45 | 15.76 | 29.27 | 44.31 | 19.03 | 11.78 | 20.20 | 33.32 | 22.73 | 13.52 | 25.47 | 41.48 |
|  | RE-NET w. attn agg. | 25.76 | 16.07 | 29.56 | 44.86 | 19.35 | 11.87 | 20.42 | 33.55 | 23.18 | 14.02 | 25.98 | 41.95 |
|  | RE-NET | 26.62 | 16.96 | 30.27 | 45.57 | 19.60 | 12.03 | 20.56 | 33.89 | 23.85 | 14.63 | 26.52 | 42.58 |
|  | RE-NET w. GT $(s, r)$ | 27.87 | 18.12 | 31.60 | 46.94 | 21.29 | 13.99 | 22.53 | 35.59 | 24.88 | 15.63 | 27.55 | 43.63 |

them as a generated graph . We set the size of entity/relation embeddings to be 200 and embedding of unobserved embeddings are randomly initialized. We use two-layer RGCN in the RGCN aggregator with block dimension $2 \times 2$. The model is trained by the Adam optimizer (Kingma & Ba, 2014). We set $\lambda_1$ to 0.1, the learning rate to 0.001 and the weight decay rate to 0.00001. All experiments were done on GeForce GTX 1080 Ti.

**Experimental Settings for Baseline Methods.** In this section, we provide detailed settings for baselines. We use implementations of TransE and DistMult[4]. We implemented TTransE and TA-DistMult based on the implementation of TransE and Distmult, respectively. For TA-DistMult, We use temporal tokens with the vocabulary of year, month and day on the ICEWS dataset and the vocabulary of year, month, day, hour and minute on the GDELT dataset. We use a margin-based ranking loss with L1 norm for TransE and use a binary cross-entropy loss for DistMult and TA-DistMult. We validate the embedding size among 100 and 200. We set the batch size to 1024, margin to 1.0, negative sampling ratio to 1, and use the Adam optimizer.

We use the implementation of ComplEx[5] Han et al. (2018). We validate the embedding size among 50, 100 and 200. The batch size is 100, the margin is 1.0, and the negative sampling ratio is 1. We use the Adagrad optimizer.

We use the implementation of HyTE[6]. We use every timestamp as a hyperplane. The embedding size is set to 128, the negative sampling ratio to 5, and margin to 1.0. We use time agnostic negative sampling (TANS) for entity prediction, and the Adam optimizer.

We use the codes for ConvE[7] and use implementation by Deep Graph Library[8]. Embedding sizes are 200 for both methods. We use 1 to all negative sampling for ConvE and use 10 negative sampling ratio for RGCN, and use the Adam optimizer for both methods. We use the codes for Know-Evolve[9]. For Know-Evolve, we fix the issue in their codes. Issues are described in Section F. We follow their default settings.

We use the code for RotatE[10]. The hidden layer/embedding size is set to 100, and batch size 256; other values follow the best values for the larger FB15K dataset configurations supplied by the author. The author reports filtered metrics only, so we added the implementation of the raw setting.

---

[4]https://github.com/jimmywangheng/knowledge_representation_pytorch

[5]https://github.com/thunlp/OpenKE

[6]https://github.com/malllabiisc/HyTE

[7]https://github.com/TimDettmers/ConvE

[8]https://github.com/dmlc/dgl/tree/master/examples/pytorch/rgcn

[9]https://github.com/rstriv/Know-Evolve

[10]https://github.com/DeepGraphLearning/KnowledgeGraphEmbedding

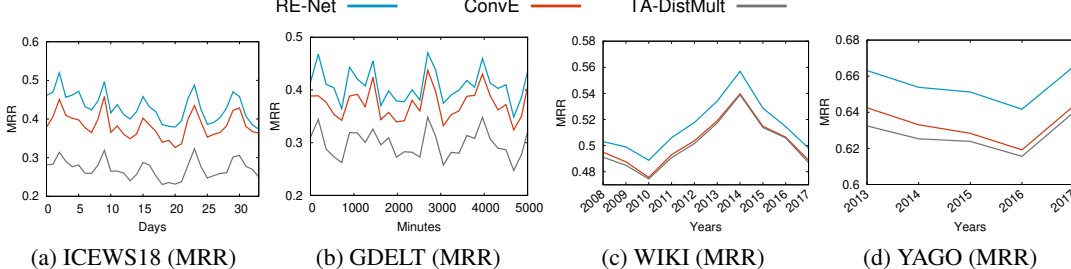

(a) ICEWS18 (MRR)    (b) GDELT (MRR)    (c) WIKI (MRR)    (d) YAGO (MRR)

Figure 6: Performance of temporal link prediction over future time stamps. We report filtered MRR (average metrics in %) on the test sets of ICEWS18, GDELT, WIKI, and YAGO datasets.

Table 5: Performance comparison on temporal link prediction on three TKG datasets with filtered metrics. RE-NET achieves the best results.

| Method | ICEWS18 - filtered | | | | ICEWS14 - filtered | | | | WIKI - filtered | | | |
|---|---|---|---|---|---|---|---|---|---|---|---|---|
| | MRR | H@1 | H@3 | H@10 | MRR | H@1 | H@3 | H@10 | MRR | H@1 | H@3 | H@10 |
| R-GCN | 23.19 | 16.36 | 25.34 | 36.48 | 26.31 | 18.23 | 30.43 | 45.34 | 37.57 | 37.44 | 39.66 | 41.90 |
| ConvE | 36.67 | 28.51 | 39.80 | 50.69 | 40.73 | 33.20 | 43.92 | 54.35 | 47.57 | 48.89 | 50.10 | 50.53 |
| ConvE+RGCN | 24.59 | 17.77 | 27.19 | 37.41 | 27.76 | 20.23 | 30.13 | 42.16 | 38.76 | 38.46 | 40.88 | 43.12 |
| EvolveRGCN | 18.50 | 16.38 | 19.19 | 22.09 | 21.70 | 18.21 | 22.81 | 28.04 | 36.58 | 32.50 | 38.10 | 41.30 |
| DynGEM | 0.00 | 0 | 0 | 0 | 0.01 | 0 | 0 | 0 | 0.01 | 0 | 0 | 0 |
| dyngraph2vecAE | 1.88 | 1.77 | 1.99 | 2.02 | 11.30 | 8.67 | 13.31 | 15.66 | 1.08 | 1.05 | 1.09 | 1.10 |
| DynTriad | 3.48 | 0 | 3.55 | 11.47 | 8.49 | 0 | 12.45 | 24.24 | 2.62 | 00.01 | 4.26 | 6.63 |
| tNodeEmbed | 8.32 | 3.19 | 9.74 | 17.47 | 17.84 | 9.98 | 20.16 | 32.88 | 9.54 | 5.78 | 10.44 | 16.60 |
| RE-NET | **42.93** | **36.19** | **45.47** | **55.80** | **45.71** | **38.42** | **49.06** | **59.12** | **51.97** | **51.01** | **52.07** | **53.91** |

# D ADDITIONAL EXPERIMENTS

## D.1 RESULTS WITH RAW METRICS.

Table 4 shows the performance comparison on ICEWS18, GDELT, ICEWS14 with raw settings. Our proposed RE-NET outperforms all other baselines. Figs. 6 shows the performance comparisons over different time stamps on the ICEWS18, GDELT, WIKI and YAGO datasets with filtered MRR. Our proposed RE-NET consistently outperform baselines over time.

## D.2 COMPARISONS WITH CONVE+RGCN.

To examine aggregation techniques in other baselines, we combine ConvE and R-GCN. We first run 2-layered R-GCN over the training graph and then each node has its own transformed representations. We run ConvE on this transformed representations. As in Table 5, ConvE+R-GCN shows better performances than R-GCN and worse performances than ConvE, which implies that the aggregation technique is not helpful to ConvE. However, aggregators in our framework is a complementary and necessary component to the temporal part, which shows superiority over the baselines.

## D.3 COMPARISONS WITH DYNAMIC METHODS.

Here we compare our method with dynamic methods on homogeneous graphs: EvolveGCN-O (Pareja et al., 2019), DynGEM (Goyal et al., 2018), dyngraph2vecAE (Goyal et al., 2019), DynTriad (Zhou et al., 2018b), and tNodeEmbed (Singer et al., 2019). These methods were proposed to predict interactions at a future time on homogeneous graphs, while our proposed method is for predicting interactions on multi-relational graphs (or knowledge graphs). Furthermore, those methods predict links at one future time stamp, whereas our method seeks to predict interactions at multiple future time stamps. We modified some methods to apply them on multi-relational graphs as follows.

**Experimental Settings.** We adopt R-GCN (Schlichtkrull et al., 2018) for EvolveGCN-O and call it EvolveRGCN. We convert knowledge graphs into homogeneous graphs for dyngraph2vecAE. The idea of this method is to reconstruct an adjacency matrix using an auto-encoder and regard it as a future adjacency matrix. If we keep relations, relation-specific adjacency matrices will be extremely sparse; the method learns to reconstruct near-zero adjacency matrices. tNodeEmbed is a temporal method on homogeneous graphs. To use this on multi-relational graphs, we first train entity

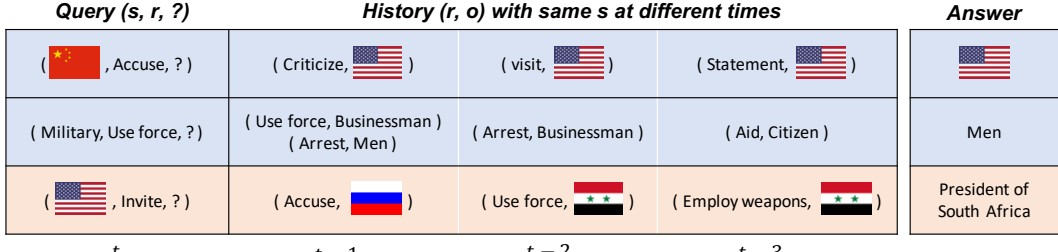

Figure 7: **Case study of RE-NET's predictions.** RE-NET's predictions depend on interaction histories. Interaction histories are categorized into three cases: (1) consistent interactions with an object, (2) a specific temporal pattern, and (3) irrelevant history. RE-NET achieves good performances on the first two cases, and poor performances on the third case.

embeddings with DistMult and set these as initial embeddings for entities in tNodeEmbed. Also we give entity embeddings as input to LSTM of tNodeEmbed. We concatenate output of LSTM and relation embeddings to predict objects. We did not modified other methods since it is not trivial to extend the methods.

**Results.** As shown in Table 5, RE-NET significantly outperforms the methods. Furthermore, all the dynamic methods do not show good performances. We conjecture that the methods are not designed for multi-relational graphs, and thus it is not capable of effectively handling multiple relations, which leads to degradation of their performances. Also, DynGEM is not suitable for our setting since it predicts edges based on observed edges at future time stamps. However, in our setting, we are not given any observed edges at future times stamps, so it shows poor performances.

## E  CASE STUDY

In this section, we study RE-NET's predictions. Its predictions depend on interaction histories. We categorize histories into three cases: (1) consistent interactions with an object, (2) a specific temporal pattern, and (3) irrelevant history. RE-NET can learn (1) and (2) cases, so it achieves high performances. For the first case, static methods cannot predict the answer since it does not see past interactions. However, RE-NET can predict the answer because it consistently interacts with an object. The second case shows specific temporal patterns on relations: ( Arrest, $o$ ) $\rightarrow$ ( Use force, $o$ ). Without knowing this pattern, one method might predict "Businessman" instead of "Men". RE-NET is able to learn these temporal patterns so it can predict the second case. Lastly, the third case shows irrelevant history to the answer and the history is not helpful to predictions. RE-NET fails to predict the third case.

## F  IMPLEMENTATION ISSUES OF KNOW-EVOLVE

We found a problematic formulation in the Know-Evolve model and codes. The intensity function (equation 3 in (Trivedi et al., 2017)) is defined as $\lambda_r^{s,r}(t|\bar{t}) = f(g_r^{s,r}(\bar{t}))(t - \bar{t})$, where $g(\cdot)$ is a score function, $t$ is current time, and $\bar{t}$ is the most recent time point when either subject or object entity was involved in an event. This intensity function is used in inference to rank entity candidates. However, they don't consider concurrent event at the same time stamps, and thus $\bar{t}$ will become $t$ after one event. For example, we have events $e_1 = (s, r, o_1, t_1), e_2 = (s, r, o_2, t_1)$. After $e_1$, $\bar{t}$ will become $t$ (subject $s$'s most recent time point), and thus the value of intensity function for $e_2$ will be 0. This is problematic in inference since if $t = \bar{t}$, then the intensity function will always be 0 regardless of entity candidates. In inference, all object candidates are ranked by the intensity function. But all intensity scores for all candidates will be 0 since $t = \bar{t}$, which means all candidates have the same 0 score. In their code, they give the highest ranks (first rank) for all entities including the ground truth object in this case. Thus, we fixed their code for a fair comparison; we give an average rank to entities who have the same scores.

## G  THEORETICAL ANALYSIS

Here we analyze the model capacity of RE-NET of capturing complex time-invariant local structure like (Hamilton et al., 2017), as well as the emerging global community structure as (You et al., 2018).

**Theorem 1** *Let $\{G_t\}_{t=1}^{\tau}$ be the snapshot of temporal knowledge graph after $\tau$ time-steps. Let $\boldsymbol{h}_v^0 \in \mathbb{R}^d, v \in \{s_i\} \cup \{o_i\}$ to be the input feature representation for Algorithm 1 of each entity node $v$. Suppose that there exists a fixed positive constant $C \in \mathbb{R}^+$ such that $||\boldsymbol{h}_v^0 - \boldsymbol{h}_{v'}^0|| > C$ for all pair of all pair of entities $v, v'$. Then we have that $\forall \epsilon > 0$, there exist a parameter setting $\Theta$ for RE-NET s.t. after $K = 4$ layers of aggregation,*

$$|\boldsymbol{h}_{v,\tau}^K - c_{v,\tau}, | < \epsilon, \forall v \in \mathcal{V}, \forall \tau \in [T],$$

*where $h_{v,\tau}^K$ are output values generated by RE-NET and $c_{v,\tau}$ are clustering coefficients of $\{G_i\}_{i=1}^{\tau}$.*

**Observation 1** *Consider a temporal graph under stochastic block model described in Section G.2. Let $h_v^0 \in \mathbb{R}^d, v \in \{s_i\} \cup \{o_i\}$ to be the input feature representation for Algorithm 1 of each node. Suppose that a constant portion $p_c$ of input representations can be linearly separated by a hyperplane, while the representation of other nodes lies on the hyperplane. There exists a parameter setting of RE-NET that can output the probability that new node $j$ connected to node $i$.*

### G.1 PROOF FOR THEOREM 1

Using pooling aggregator of GraphSAGE, we can actually copy its behavior by setting recurrent weight matrix of the RNN model to be $0$. In this case, we lose all time-dependency our RE-NET and the representation model becomes *time-invariant*. However, RE-NET have exactly the same model capacity as GraphSAGE.

### G.2 ANALYSIS FOR OBSERVATION 1

Here we define the generation process of our temporal graph. Assume that the generation process of the graph follows a stochastic block model, and there are two communities in the graph. Half of the nodes belong to community A and the other half belong to community B. Nodes within one community have probability $p_s$ to be connected while other pairs have $p_d < p_s$ probability to be connected. The edges in the graph are introduced into the graph in a time. Suppose a sequence of time-steps, a new node is introduced to the community and each edge is added to the graph.

This observation follows from three facts: (1) For each node $v_j$ in the neighborhood $\mathcal{N}(v)$, using pooling aggregator, we can detect their community assignment $s_j$. We assign the output of community A to be $+1$ and the output of community B to be $-1$. (2) The error of incorrectly discerning the community of a node decrease exponentially with the number of links. For example let the node $v$ be in community A. Let the total number of nodes at time $t$ to be $n_t$, by Hoeffding's inequality we have

$$\mathbb{P}\left(\sum_{j:v_j \in \mathcal{N}(j)^t} s_j < 0\right) < exp(-2(p_s - p_d)^2 |\mathcal{N}(j)^t|)$$

(3) Given the correct community classification, the relation classifier is able to predict the probability of linking nodes.

Combining these three facts, RE-NET is able to infer the community structure of the node.

