# OpenReview forum: "Recurrent Event Network : Global Structure Inference Over Temporal Knowledge Graph"
_ICLR.cc/2020/Conference — Reject_

### Official Review · AnonReviewer1 · 2019-10-23
**Official Blind Review #1**

**Rating:** 3

**Review:**

This paper properly applied several technique from RNN and graph neural networks to model dynamically-evolving, multi-relational graph data. There are two key component: a RNN to encode temporal information from the past event sequences, and a neighborhood aggregator collects the information from the neighbor nodes. The contribution on RNN part is design the loss and parameterizes the tuple of the graph. The contribution of the second part was adapting Multi-Relational Aggregator to this network. The paper is well-written. Although I'm familiar with the dataset, the analysis and comparison seems thorough.

I'm leaning to reject or give borderline for this paper because (1) This paper is more like an application paper. Although the two component is carefully designed, the are more like direct application. I'm not challenge this paper is not good for the target task. But from the point of view on Machine learning / deep learning, there is not much insight from it. The technical difficult was more from how to make existing technique to fit this new problem.  This "new" problem seems more fit to data mining conference. (2) The experiments give tons of number but it lack of detailed analysis, like specific win/loss case of this model. As a more application-side paper, these concrete example can help the reader understand why this design outperform others. For example, it can show what the attention weights look like, and compare to the proposed aggregator.

Some questions:
[This question is directly related to my decision] Does this the first paper to apply autoregressive to knowledge graph? from related work, the answer is no. Can the author clarify more on this sentence?

"In contrast, our proposed method, RE-NET, augments a RNN with message passing procedure between entity neighborhood to encode temporal dependency between (concurrent) events (i.e., entity interactions), instead of using the RNN to memorize historical information about
the node representations."

The paper give complexity of this algorithm but no comments about how it compare with other method and how practical it is.

It lacks of some details for the model:
(1) what is the RNN structure?
(2) For the aggregator, what is the detailed formulation of h_o^0?


**Experience Assessment:**

I have read many papers in this area.

**Review Assessment: Checking Correctness Of Derivations And Theory:**

I carefully checked the derivations and theory.

**Review Assessment: Checking Correctness Of Experiments:**

I assessed the sensibility of the experiments.

**Review Assessment: Thoroughness In Paper Reading:**

I read the paper thoroughly.

---

> ### Author Response · Authors · 2019-11-12
> **Response to reviewer 1 [3/3]**
>
> Q4: Regarding “The paper lacks detailed analysis.”
>
> A4: Thank you for pointing out this place to improve! First, we would like to point out that in prior submission we conducted analysis and ablation studies on model components and model variants (in Section 4.3 and 4.4) to help readers understand which parts of the proposed model are useful and how important they are.
>
> Moreover, with limited rebuttal time, we added case study about RE-Net’s predictions in Section E (Appendix). RE-Net’s predictions depend on interaction histories. The histories can be categorized into three cases: (1) consistent interactions with an object, (2) a specific temporal pattern, and (3) irrelevant history. In the first case, history has interactions with the same objects but with different relations. In the second case, history shows a specific temporal pattern on relations such as ( s, Arrest, o) -> ( s, Use force, o). In the third case, there are no relevant interaction histories to the query. RE-Net learns (1) and (2) cases, so it achieves good performances. However, RE-Net cannot predict answers given irrelevant history (case (3)). Please refer to Section E for details.
>
> -------------------------------
> Q5: Regarding “The paper give complexity of this algorithm but no comments about how it compare with other method and how practical it is.”
>
> A5: We appreciate the reviewer’s careful comments. We have updated time complexity for generating one graph in Section 3.3. The time complexity is linear to the number of entities and relations. Other methods do not generate graphs and do not perform multi-step predictions, whereas our method do with Algorithm 1. Thus we couldn’t compare our time complexity with other methods. RE-Net have a limitation on generating huge graphs which contain billions of nodes. We left the work of designing an efficient generative model of TKGs as future work.
>
> -------------------------------
> Q6: What is the RNN structure?
>
> A6: Thanks for pointing out and sorry for the confusion! We adopt Gated Recurrent Units as our RNN structure in the experiments. The definition of RNN was given in appendix A in previous submission. We also added a short description of the RNN structure in Section 3.1.
>
> -------------------------------
> Q7: For the aggregator, what is the detailed formulation of h_o^0?
>
> A7: Thanks for pointing out the confusion about the initial hidden representation! We set the initial hidden representations of each node (h_o^0) in the RGCN aggregator as trainable embedding vectors of each node (e_o). We added details on this in section 3.2 of the updated paper.

---

> ### Author Response · Authors · 2019-11-12
> **Response to reviewer 1 [2/3]**
>
> Q2: From related work, the answer to “Is this the first paper to apply autoregressive to knowledge graphs?” is no. Can the author clarify more on the following sentence? —>  "In contrast, our proposed method, RE-NET, augments a RNN with message passing procedure between entity neighborhood to encode temporal dependency between (concurrent) events (i.e., entity interactions), instead of using the RNN to memorize historical information about the node representations."
>
> A2: Sorry for the confusion caused by the above sentence in our related work! We meant to say that the usage of RNN (i.e., the idea of autoregressive modeling) is rather different from previous work which also leverage RNNs in their model architectures, such as EvolveGCN [1] and GCRN [2].
>
> RNNs in our work are used to implement the recurrent event encoder (Eqs. (3)-(8) in the paper), which aims to parameterize the joint probability distribution for P(s, r, o | G_<t), i.e., modeling the sequence of graph snapshots {G_t} (events) in an autoregressive fashion.
>
> In contrast, prior work adopt RNN to either memorize and update the states of node embeddings that are dynamically evolving (e.g., in GCRN [2]), or memorize and update the GCN model parameters for different time stamps (e.g., in EvolveGCN [1]). While our work aims to leverage autoregressive modeling to formulate the structure inference problem for temporal KGs, previous work has not applied RNNs to predict event sequence and use RNNs in rather different ways.
>
> We have updated our related work section to clarify this question. Thank you!
>
> [1] Evolve-GCN: Evolving Graph Convolutional Networks for Dynamic Graphs, Pareja et. al.
> [2] Structured Sequence Modeling with Graph Convolutional Recurrent Networks, Seo et al.
>
> -------------------------------
> Q3: Regarding “This paper is more like an application paper. From the point of view on Machine learning / deep learning, there is not much insight from it.”
>
> A3: We very much appreciate the reviewer’s comments. Designing generative models for sequential, event data is an important topic in machine learning. Compared with the problems studied in existing work (e.g., graph generation, text generation), our problem is more challenging, as (1) the data instance at each time step is a multi-relational graph, (2) the structure of which is very complicated, and modeling such highly complicated data is nontrivial.
>
> Regarding (1), our problem on multi-relational graphs (knowledge graphs) is more challenging than on homogeneous graphs since knowledge graphs have multi-relational edges. This leads to a huge search space ( (# of relations) times bigger) and needs more sophisticated modeling. To tackle this problem, we design the joint probability of a knowledge graph as discussed in A1.
>
> Regarding (2), the structure of multi-relational graphs has its complex nature. Modeling such a multi-relational graph in an autoregressive manner requires big contributions to effectively include the structure of multi-relational graphs as discussed in A1. To tackle this, we propose our recurrent event encoder to parametrize the probability function for P(s, r, o | G_<t) with aggregators.
>
> Furthermore, our paper is the first work that proposes an autoregressive-based generative model for this problem. In this sense, our work still makes many contributions to the machine learning field.

---

> ### Author Response · Authors · 2019-11-12
> **Response to reviewer 1 [1/3]**
>
> Thank you very much for the helpful feedback and comments!
>
> We did our best to address the raised questions and incorporated the comments in our updated draft. In particular, we have included case study on concrete examples in Section E of the Appendix to help understand how our method work, and also updated the writing of complexity analysis in Section 3.3 to clarify things.
>
> Below please find our detailed responses to the questions in your review. We sincerely hope you could check our response (especially Q1-Q4), and consider changing the evaluation on our paper!
>
> -------------------------------
> Q1: Regarding “Is this the first paper to apply autoregressive to knowledge graphs?” and our technical contributions.
>
> Yes. To the best of our knowledge, this is the *first work* to formulate the structure inference (prediction) problem for temporal, multi-relational (knowledge) graphs *in an autoregressive fashion*—-i.e., aiming to model and predict the occurrences (events) of entities and their relationships in the next (future) time stamp based on priorly observed events.
>
> Our autoregressive problem formulation is new and technically challenging *particularly in the context of multi-relational (knowledge) graph* because: (1) one needs to consider the interdependence between different entities (e.g., some events involving the same entities are more likely to happen together than others) rather than modeling the event sequence for each entity, e.g., [(e, r1, o1; t1), (e, r2, o2; t2), (e, r3, o3; t3), ….], in an independent manner; and (2) one needs to carefully model the probability of a triple (event) given the history, i.e., P(s, r, o | G_<t), which requires both *effective and tractable* solution to deal with multi-relational data—-e.g., directly parametrizing P(s, r, o | G_<t) will result in a huge output space that is unscalable and difficult to learn.
>
> Regarding challenge (1) above, we propose to impose conditional independence over the events at the same time stamp (i.e., Eq (1) in the paper), where events in G_t are mutually independent given the previous events G_t-m:t-1. We also define the joint probability distribution of an event based on past events and the involved entities (i.e., Eq (2) in the paper). This ensures that the prediction of a new event will be made based on events of all the entities in previous time stamps (i.e., to model the interdependence). The formulation of this joint probability distribution is new and has not been studied in previous work.
>
> Regarding challenge (2), we propose our recurrent event encoder (Eqs. (3)-(8) in the paper) to parametrize the probability function for P(s, r, o | G_<t). In particular, multi-relational data has a much larger search space due to its complex nature, i.e., |entities|-by-|relations|-by-|entities|. Thus, we need a careful design of the encoder to ensure the *tractability*. We have put significant efforts on exploring the parameterization design, and finally reached to the current formulation of p(s|G_<t), P(r | s, G_<t) and P(o | s, r, G_<t).
>
> To our knowledge, the most related work in terms of autoregressive modeling of graphs is GraphRNN [1]. However, we would like to point out that GraphRNN focuses on a rather different problem setting: their input is a set of static, single-relational (homogeneous) graphs from the same distribution and their goal is to learn a generative model for this family of graphs (i.e., the data distribution) so as to generate more graphs of the same kind. The main challenge resolved in GraphRNN is how to deal with permutation-invariance of static graphs and how to learn the generative model from many graph examples, which are different from our focus in this paper. We updated our related work section to stress on this.
>
> We make sure these points are well stressed in the revised version.
>
> [1] Graphrnn: Generating realistic graphs with deep auto-regressive models, You et al., ICML 2018

---

### Official Review · AnonReviewer2 · 2019-10-23
**Official Blind Review #2**

**Rating:** 6

**Review:**

This paper presents the Re-Net model which sequentially generates a temporal knowledge graph (TKG) in an autoregressive fashion by taking both global and local information into account.
The generation of TKG is motivated via a joint distribution problem which is then parametrized by the usage of a recurrent event encoder.
In addition to past information, the encoder aggregates local as well as global information for which the authors propose three different aggregation schemes build upon the works of, e.g., attentive pooling and the RGCN model.
In an in-depth-evaluation study, the performance of the proposed model is evaluated on five different datasets on which it consistently improves upon the state-of-the-art.
An ablation study shows the benefits of all proposed features of Re-Net, e.g., the usage of more sophisticated aggregation schemes, the impact of using global information, and the number of RGCN layers.

As far as I know, the proposed method is a novel and clever (though not ground-breaking) contribution to the field of performing global structure inference over TGKs.
The paper is well-written but is partially becoming a little hard to comprehend due to its overloaded notation, e.g., $h_t(s)$ vs. $h_s^l$ and $N(s)_t$ vs. $N_t^{(s)}$ vs. $N_t^{(s,r)}$, and could be improved by a more rigorous formulation, e.g., for $N(s)_t$ or $c_s$ (which should also depend on r).

1. Re-Net evolves the embeddings of entities and performs predictions via negative log likelihood. Hence, the model seems to be limited to predict events between entities which have been already seen during training and does not generalize to unseen entities. In addition, by applying a softmax classifier your model does not seem to be able to scale to large knowledge graphs? Are those observations correct and how could they be resolved?

2. As far as I understood, the formulation of $N^{(s,r)}$ is not needed for defining the mean and attentive pooling aggregators since you are aggregating information independent of the relation type. However, the current formulation could confuse readers (including me).

3. Algorithm 1 could be made more clear since the sampled number of M subjects does not get mentioned again. I guess the top-k triples are picked across all M samples and not individually? In addition, the sampling of subjects should relate to Equation 5 instead of Equation 4.

4. In Figure 3 it is not clear why the accuracy does not decrease with temporal distance from the training examples. It would be helpful to interpret and clarify the results in more detail.

5. I was not able to fully comprehend your complexity analysis. For example, it is not clear what $|E|$ means (I guess the maximum number of triples in a time step?). In addition, it seems that you are still dependent on computing node embeddings for all entities in your graph, even if you only report runtimes for computing a single example. In my opinion, there is a $L \cdot |E|$ term missing in your complexity analysis for computing RGCN across the whole graph. Please clarify!

6. The results of using the attentive aggregation scheme should be included into Tables 1 and 2.

7. Since Figure 5c signalizes that Re-Net can effectively leverage larger receptive field sizes, how does it perform when increasing the number of layers further?

------------------------
Update after the rebuttal:

I would like to thank the authors for answering my questions and clarifying several issues. The raised questions were not critical for my overall rating, which remains unchanged (6: Weak Accept).

**Experience Assessment:**

I have read many papers in this area.

**Review Assessment: Checking Correctness Of Derivations And Theory:**

I assessed the sensibility of the derivations and theory.

**Review Assessment: Checking Correctness Of Experiments:**

I assessed the sensibility of the experiments.

**Review Assessment: Thoroughness In Paper Reading:**

I read the paper at least twice and used my best judgement in assessing the paper.

---

> ### Author Response · Authors · 2019-11-11
> **Response to reviewer 2 [2/2]**
>
> Q6: Regarding “In Figure 3 it is not clear why the accuracy does not decrease with temporal distance from the training examples.”
>
> A6: The reviewer pointed out that in Figure 3 it is not clear why the accuracy does not decrease with temporal distance from the training examples. The reason why the performances fluctuated is because testing entities are not always the same entities at each time.  We can verify this from ConvE’s performances over time.  ConvE is a static method which is not affected by temporal distances. ConvE’s accuracy does not decrease in the Figure. In other words, testing entities are different at each time step, thus leading to fluctuation. Instead, we have to focus on the difference between RE-Net and ConvE methods. The difference gets smaller as time increases. We added this in Section 4.2.
>
> -------------------------------
> Q7: Regarding “I was not able to fully comprehend your complexity analysis.  There is a term L * |E| missing in your complexity analysis for computing RGCN across the whole graph”
>
> A7: Sorry for the confusion. We have updated complexity analysis for generating one graph in Section 3.3. We provide a more detailed analysis. As reviewer said, |E| means the maximum number of triples in a time step. We have added L*|E| term which is needed to compute P(s|G_t-m:t-1). The time complexity of RE-Net is linear to |E|, number of entities and relations. Thanks for pointing out!
>
> -------------------------------
> Q8: Regarding “The results of using the attentive aggregation scheme should be included into Tables 1 and 2.”
>
> A8: We have updated tables 1, 2, and 4 to include RE-Net with an attentive aggregator. It shows improvements over the mean aggregator, which implies that giving different attention weights to each neighbor helps predictions.
>
> Summarized results on ICEWS18 are as follows:
>
> Method		                | MRR  | Hits@1 | Hits@3 | Hits@10
> —————————————————————————--
> RE-Net w. mean agg.	| 40.70 | 34.24    | 43.27    | 53.65
> RE-Net w. attn agg.	| 40.96 | 34.57    | 44.08    | 54.32
> RE-Net		                | 42.93 | 36.19    | 45.47    | 55.80
>
> -------------------------------
> Q9: How does it perform when increasing the number of layers further in Figure 5c?
>
> A9: We have added a result of the 3-layered model. It underperforms 2-layered model. We conjecture that the bigger parameter space leads to overfitting.

---

> ### Author Response · Authors · 2019-11-11
> **Response to reviewer 2 [1/2]**
>
> We thank for positive feedback and thoughtful suggestions. We tried to resolve the issues and incorporated the comments in our updated draft. We also revised writing about notations and complexity analysis, etc, and updated experiments in Section 4.
>
> -------------------------------
> Q1: Regarding “The paper is a little hard to comprehend due to its overloaded notations.”
>
> A1: Thanks for suggesting notations! We have updated the notations in the paper. We mostly used the same notations since we think current notations are intuitive. We tried to remove the confusion. The definition of each notation is as follows:
> N_t^(s) : neighbors of s at time t.
> N_t^(s,r) : neighbors of s under relation r at time t.
> h_t(s) : a history vector of s at time t.
> h_s^(l) : a hidden representation of RGCN (needs update).
> Note that h_t(s) and h_s^(l) are different notations. We use different bold faces to differentiate them.
>
> -------------------------------
> Q2: Regarding “The model seems to be limited to predict events between entities which have been already seen during training and does not generalize to unseen entities.”
>
> A2: Thanks for the great question. As the reviewer said, RE-Net does not generalize to unseen entities in the current setting. This is because the entity attributes are not given in the dataset. However, If we have entity attributes for training and test sets, RE-Net can generalize to unseen entities by using entity attributes. For example, embeddings for each node will be [attributes] * W. W is a trainable matrix. In the current setting, We defined embeddings as [one-hot vector] * W. By using one-hot vector, parameters for unseen entities are not learned during training.
>
> -------------------------------
> Q3: Regarding “By applying a softmax classifier your model does not seem to be able to scale to large knowledge graphs.”
>
> A3: Yes, I agree that a softmax classifier does not scale. To deal with huge graphs, we can adopt a sigmoid function with a negative sample strategy instead of a softmax function. Then, it can scale to large knowledge graphs.
>
> -------------------------------
> Q4: Regarding “The formulation of is not needed for defining the mean and attentive pooling aggregators since you are aggregating information independent of the relation type.”
>
> A4: Thanks for the comment! The reviewer commented that the formulation is not needed for defining the mean and attentive pooling aggregators since RE-Net aggregates information independent of the relation type. However, mean and attentive pooling aggregators adopt a different strategy from a multi-relational aggregator. Since the mean and attentive pooling aggregator does not deal with multiple relations, they only aggregate neighbors under the same and fixed relation. For example, if we want to get P(o|s,r,N), then the aggregator only collects neighbors under relation r. We added the description in Section 3.2 of the paper.
>
> -------------------------------
> Q5: The top-k triples are picked across all M samples and not individually? And Equation 4 should be Equation 4 in Algorithm 1.
>
> A5: Thanks for pointing out the typo! As the reviewer said, top-k triples are picked across all M samples. We have updated writing in Section 3.3. We also have fixed the typo about Equation 4.

---

### Official Review · AnonReviewer3 · 2019-11-04
**Official Blind Review #3**

**Rating:** 3

**Review:**

The paper proposes a recurrent and autorgressive architecture to model temporal knowledge graphs and perform multi-time-step inference in the form of future link prediction. Specifically, given a historical sequence of graphs at discrete time points, the authors build sequential probabilistic approach to infer the next graph using joint over all previous graphs factorized into conditional distributions of subject, relation and the objects. The model is parameterized by a recurrent architecture that employs a multi-step aggregation to capture information within the graph at particular time step. The authors also propose a sequential approach to perform multi-step inference. The proposed method is evaluated on the task of future link prediction across several baselines, both static and dynamic, and ablation analysis is provided to measure the effect of each component in the architecture.

The authors propose to model temporal knowledge graphs with the key contribution being the sequential inference and augmentation of RNN with multi-step aggregation. The paper is well written in most parts and provides adequate details with some exceptions. I appreciate the extended ablation analysis as it helps to segregate the effect of each component very clearly. However, there are several major concerns which makes this paper weaker:

- The paper approaches temporal knowledge graphs in discrete-time fashion where multiple events/edges are available at each time step. While this is intuitive, the authors fail to position the paper in light of various existing discrete-time approaches that focus on representation learning over evolving graphs [1,2,3,4,5]. Related work mentions [1] learns evolving representations but all these methods can do future link prediction and hence this is a big miss for the paper. A discussion and comparison with these approaches is certainly required as most of static and dynamic baselines currently compared also focus on learning representations, hence that is not a valid argument to miss comparison.

- The baselines tested by the authors are either support static graphs, supports interpolation or supports continuous time data. However, as the authors explicitly propose a discrete time model starting from Section 3, it is important to perform experiments on atleast few of the discrete time baselines to demonstrate the efficacy of the proposed method. For instance, authors can augment relation as extra feature or use their encoders and optimization function to perform experiments e.g. Evolve-GCN  only require to replace GCN with R-GCN.

- From the ablation it is clear that aggregation is the most important component as without it, the performance drops much closer to ConvE which is a static baseline and significantly worse than other RE-Net variants. However, the aggregation techniques are not novel contributions but augmentation to the RNN architecture. Hence it is important to show how augmenting aggregation module with other baselines (for instance, ConvE and TA-DistMult)) and the above mentioned discrete baselines would affect the performance of these baselines.

- While the authors describe attentive Pooling Aggregator, the experiments only show mean aggregator and multi-step one. Is there a reason Attentive pooling is not used for any experiments?
-It appears that global vector H_t is not playing significant role based on ablation study. Can the authors explain why that si the case? Also, what aggregation is used to compute H_t? Is it sum over all previous h_t's?

- Algorithm 1 is not very clearly explained. When the authors mention that they only use one sample, does that mean a single subject is sampled at each time point t'? If so, how do you ensure the coverage is good across subjects in the newly generated graph? I admit I am not clear on this and would recommend the authors to elaborate in response and also in the paper. Also, the inference computation complexity is concerning. While it seems fine for the provided dataset, most real-world graphs have billion of nodes and I all of E, L and D would be larger for such graphs. This seems to put a strict limitation on scalability of inference module.

- It is not clear what is the difference between RE-NET and RE-NET w. GT. Could the authors elaborate this more? It seems the authors do not update history when they perform RE-NET w/o multi-step. However, in the RE-NET w. GT, where is the ground truth history used in Algorithm 1?

- The time span expansion for WIKI and YAGO is very unnatural and it is not clear if these experiments provide any value. For instance, can the authors show that in multi-step inference scheme, they can actually predict events at multiple time points corresponding to time span events in actual dataset? As multiple triplets can appear at consecutive time points, the current modification just makes them equivalent which doesn't seem correct.

I am willing to revisit my score if the above concerns are appropriately addressed and requested experiments are provided.

[1] Evolve-GCN: Evolving Graph Convolutional Networks for Dynamic Graphs, Pareja et. al.
[2] DynGEM: Deep embedding method for dynamic graphs, Goyal et. al.
[3] dyngraph2vec: Capturing network dynamics using dynamic graph representation learning, Goyal et. al.
[4] Dynamic Network Embedding by Modeling Triadic Closure Process, Zhou et. al.
[5] Node Embedding over Temporal Graphs, Singer et. al.

**Experience Assessment:**

I have published one or two papers in this area.

**Review Assessment: Checking Correctness Of Derivations And Theory:**

N/A

**Review Assessment: Checking Correctness Of Experiments:**

I carefully checked the experiments.

**Review Assessment: Thoroughness In Paper Reading:**

I read the paper thoroughly.

---

> ### Author Response · Authors · 2019-11-11
> **Response to reviewer 3 [3/3]**
>
> Q9: Regarding “It is not clear what is the difference between RE-NET and RE-NET w. GT.”
>
> A9: As the reviewer said, RE-Net w. GT does not update history (or generate a graph) since it already has ground truth history (“Variant of RE-Net” in Section 4.1). Thus it does not need to use Algorithm 1. In this case, \hat{G}_{t+1:t+\Delta t -1} is known from ground truth history. For inference, it uses equation (3) which is a probability for o_t given s, r, and history. Since RE-Net does not know ground truth, it needs to generate history of triples (or a graph) which is described in Algorithm 1. We have updated writing in Section 4.1.
>
> -------------------------------
> Q10: Regarding “The time span expansion for WIKI and YAGO is very unnatural and it is not clear if these experiments provide any value.”
>
> A10: Thanks for the great question. WIKI and YAGO datasets are different from ICEWS and GDELT. TA-DistMult [1] and HyTE [2] also used these two datasets. WIKI and YAGO are not event-based but have time-ranged facts. In other words, WIKI and YAGO datasets have temporally associated facts (s,r,o,t1,t2) where t1 means a starting time and t2 means an ending time. We should convert them into an event-based setting (s,r,o,t1), (s,r,o,t1+1), ..., (s,r,o,t2) to do multi-step inference, since RE-Net takes sequences of triples and predicts interactions in a discrete manner. I agree that this conversion can be unnatural. We leave the sophisticated modeling for future work. Thanks for the sharp question!
>
> [1] Learning Sequence Encoders for Temporal Knowledge Graph Completion, García-Durán et al.
> [2] HyTE: Hyperplane-based Temporally aware Knowledge Graph Embedding, Dasgupta et al.

---

> ### Author Response · Authors · 2019-11-11
> **Response to reviewer 3 [2/3]**
>
> Q3: Regarding “However, the aggregation techniques are not novel contributions but augmentation to the RNN architecture. Hence it is important to show how augmenting aggregation module with other baselines and the above mentioned discrete baselines would affect the performance of these baselines.”
>
> A3: Thank you for making a good point to strengthen our experiment. We agree that the aggregation methods adopted in our model are not new and have been used in prior work. However, we would like to stress that the integration of two complementary modules (1) autoregressive modeling of the temporal aspect of the data (i.e., event sequences) and (2) aggregation over the local structural neighborhood (i.e., concurrent events within a time window that form a sub-graph) is novel and is one of our contributions. Such an integration of the two mutually enhancing modules has not been well understood. However, we argue that integrating aggregation module with static methods (e.g., ConvE) or temporal baselines that are also exploring structural similarity/consistency (e.g., DynTriad, DynGem, and TA-TransE/DistMult) may not bring significant gains, as the aggregation module also exploits the structural information in the local neighborhood.
>
> With limited response time, we conduct experiments on ConvE+RGCN over the ICEWS18 and summarize the results as follows. ConvE+R-GCN shows better performances than R-GCN and worse performances than ConvE, which implies that the aggregation technique is not helpful to ConvE.
>
> Method		        | MRR  | Hits@1 | Hits@3 | Hits@10
> —————————————————————————--
> ConvE+RGCN	| 24.59 | 17.77    | 27.19    | 37.41
> RGCN			| 23.19 | 16.36    | 25.34    | 36.48
> ConvE			| 36.67 | 28.51    | 39.80    | 50.69
> RE-Net		        | 42.93 | 36.19    | 45.47    | 55.80
>
> -------------------------------
> Q4: Regarding “Results of RE-Net with an attentive aggregator are missing.”
>
> A4: Thank you for asking additional comparison to help validate our claims. We have updated tables 1, 2 & 4 to include an additional model variant, “RE-Net with attentive aggregator”. It shows improvements over the mean aggregator, which implies that giving different attention weights to each neighbor helps predictions. In particular, results on ICEWS18 are summarized as follows.
>
> Method		                | MRR  | Hits@1 | Hits@3 | Hits@10
> —————————————————————————--
> RE-Net w. mean agg.	| 40.70 | 34.24    | 43.27    | 53.65
> RE-Net w. attn agg.	| 40.96 | 34.57    | 44.08    | 54.32
> RE-Net		                | 42.93 | 36.19    | 45.47    | 55.80
>
> -------------------------------
> Q5: Can the authors explain why the global vector H_t is not playing a significant role?
>
> A5: Thanks for the great question. H_t does not play a significant role in performances since most important information comes from local neighborhood. For example, most interactions occur between local neighborhood not distant neighbors. Global information H_t serves as a complement of local neighborhood and give information about distant neighbors.
>
> -------------------------------
> Q6: What aggregation is used to compute H_t?
>
> A6: Thanks for pointing out and sorry for the confusion. We use the multi-relational aggregator to compute H_t. After going through multi-relational aggregator, each node has h_t. We max-pool all h_t and this will become H_t. We have updated writing of “Recurrent Event Encoder” in Section 3.1.
>
> -------------------------------
> Q7: Algorithm 1 is not very clearly explained. When the authors mention that they only use one sample, does that mean a single subject is sampled at each time point t'?
>
> A7: Thanks for pointing out the confusion about sampling process. We sample one graph in Algorithm 1. To get one graph at each time step, we first sample M number of subjects and pick top-k triples which are from Equation 2. Then, we have k triples which build a temporal knowledge graph at time t. We have updated writing at the end of “Multi-step Inference over Time” in Section 3.3.
>
> -------------------------------
> Q8: Regarding “Inference computation complexity is concerning. There is a strict limitation on scalability of inference module.”
>
> A8: This is a great question. As the reviewer said, RE-Net has some limitation on scalability. The time complexity for generating one graph is linear to the number of entities and relations. We have updated time complexity in Section 3.3. If the graph has billions of nodes, we need a more efficient version of RE-Net. We didn’t proposed the efficient version. We leave this for future work.

---

> ### Author Response · Authors · 2019-11-11
> **Response to reviewer 3 [1/3]**
>
> Thank you very much for your helpful feedback and valuable comments! We appreciate the questions you raised regarding the missing related work/baselines and analysis on model components. We answer each question in the response below and have revised our draft to incorporate the comments. In particular, we have added comparison with the suggested baselines in the appendix (Section D) and explain our method’s advantages over them. We hope the reviewer can check our response and new experiments, and consider changing the evaluation of our paper!
>
> -------------------------------
> Q1: Comparison with other existing discrete-time approaches that focus on representation learning over evolving graphs  [1,2,3,4,5].
>
> A1: Thanks for the suggestion! The main reasons why we didn’t include detailed discussion on these recent work on learning representation for dynamic graphs (with discrete time stamps) are: (1) They study “single-relational” (homogeneous) graphs while our work focuses on “multiple-relational” knowledge graphs; and (2) these methods are designed to predict future relationship “in one step” (i.e., for t+1), while our method aims for multi-step prediction (i.e., for t+1, t+2, t+3, …, t+k). We will expand our related work section to include a more detailed discussion on these methods.
>
> Regarding (1), one can change the GCN in [1] into a R-GCN to deal with multi-relational graphs but it is non-trivial to modify [2-4] to extend to multi-relational graphs. We added comparison to a new baseline, EvolveRGCN, by replacing GCN in Evolve-GCN [1] by a R-GCN, and also compare with DynGEM [2], dyngraph2vec [3], DynTriad [4] by ignoring the relation type information on the edges (i.e., changing the knowledge graph to single-relational graph). Also, we add relation embeddings for tNodeEmbed [5] so it can be applied to multi-relational graphs. Here, we briefly summarize the results on the ICEWS18 dataset as follows. More detailed results (over three datasets, all metrics) can be found in Section D.3 in the Appendix of the updated draft.
>
> Method		        | MRR  | Hits@1 | Hits@3 | Hits@10
> —————————————————————————--
> EvolveRGCN		| 18.50 | 16.38    | 19.19    | 22.09
> DynGEM		| 0.00   | 0            | 0           | 0
> dyngraph2vecAE	| 1.88   | 1.77      | 1.99      | 2.02
> DynTriad		| 3.48   | 0           | 3.55      | 11.47
> tNodeEmbed	| 8.32   | 3.19      | 9.74      | 17.47
> RE-Net		        | 42.93 | 36.19    | 45.47    | 55.80
>
> We can see that RE-Net outperform these five (modified) baselines by a large margin. In fact, these single-relational, one-step prediction models cannot outperform some strong static graph baselines like ConvE. We conjecture that the methods are not specifically designed for multi-relational graphs, and thus they are not capable of handling multiple relations, which leads to degradation of their performances.
>
> Regarding (2), the suggested methods may be tweaked to predicted G_{t+2} if their models can be re-trained over G_{1, .., t, t+1}. However, such re-training is time consuming and relies on “predicted labels” G_{t+1}. Due to limited time for rebuttal, we leave the in-depth analysis of this problem to future work.
>
> [1] Evolve-GCN: Evolving Graph Convolutional Networks for Dynamic Graphs, Pareja et. al.
> [2] DynGEM: Deep embedding method for dynamic graphs, Goyal et. al.
> [3] dyngraph2vec: Capturing network dynamics using dynamic graph representation learning, Goyal et. al.
> [4] Dynamic Network Embedding by Modeling Triadic Closure Process, Zhou et. al.
> [5] Node Embedding over Temporal Graphs, Singer et. al.
>
> -------------------------------
> Q2: Regarding “The baselines tested by the authors are either support static graphs, supports interpolation or supports continuous time data.”
>
> A2: Thanks for the suggestion on strengthening our experiments. In our prior submission, we modified DyRep [1] (into DyRep+MLP) and GCRN [2] (into R-GCRN+MLP) as baselines, which deal with extrapolation, discrete-time, and multi-relational temporal graph prediction, same as the setting of our focus. The experiments show that RE-Net significantly outperforms these methods. We will include more discussion on other recent related work as mentioned in Q1. We have updated our draft to include other methods as discussed in our response A1.
>
> [1] Dyrep: Learning representations over dynamic graphs, Trivedi et al.
> [2] Structured Sequence Modeling with Graph Convolutional Recurrent Networks, Seo et al.

---

### Author Response · Authors · 2019-11-11
**Paper revision**

We would like to thank all the reviewers for their thoughtful and valuable comments. We have carefully revised the paper to address the comments from reviewers, and also added additional experiments to support our method. The major updates of the draft are listed as follows:

- Added more related works in Section 2, as suggested by reviewers 1 and 3.
- Updated complexity analysis in Section 3.3, as suggested by reviewers 1, 2, and 3
- Added experiments of RE-Net with an attentive aggregator in Table 1,2, and 4, and other baselines in Section D, as suggested by reviewers 2 and 3
- Added case study of RE-Net in Section E, as suggested by reviewer 1.
- Improved clarity of Sections 3 and 4 to reflect the questions raised by reviewers.

---

### Decision · Program_Chairs · 2019-12-19

**Decision:**

Reject

**Comment:**

The paper proposes a recurrent and autorgressive architecture to model temporal knowledge graphs and perform multi-time-step inference in the form of future link prediction. However, the reviewers feel that the papers are more of a straight application of current techniques. Furthermore, a better presentation of the experimental section will also help improve the paper.